# Structural mechanism of FusB-mediated rescue from fusidic acid inhibition of protein synthesis

Adrián González-López [1,2], Xueliang Ge [1], Daniel S. D. Larsson [1],
Carina Sihlbom Wallem[3], Suparna Sanyal [1] & Maria Selmer [1,2] ✉

The antibiotic resistance protein FusB rescues protein synthesis from inhibition by fusidic acid (FA), which locks elongation factor G (EF-G) to the ribosome after GTP hydrolysis. Here, we present time-resolved single–particle cryo-EM structures explaining the mechanism of FusB-mediated rescue. FusB binds to the FA-trapped EF-G on the ribosome, causing large-scale conformational changes of EF-G that break interactions with the ribosome, tRNA, and mRNA. This leads to dissociation of EF-G from the ribosome, followed by FA release. We also observe two independent binding sites of FusB on the classical-state ribosome, overlapping with the binding site of EF-G to each of the ribosomal subunits, yet not inhibiting tRNA delivery. The affinity of FusB to the ribosome and the concentration of FusB in *S. aureus* during FusB-mediated resistance support that direct binding of FusB to ribosomes could occur in the cell. Our results reveal an intricate resistance mechanism involving specific interactions of FusB with both EF-G and the ribosome, and a non-canonical release pathway of EF-G.

Antibiotic resistance is mediated by a range of different mechanisms, including drug efflux, mutations or modifications of the target, degradation or modification of the drug, and target protection[1]. Of these, target protection, involving a resistance protein that provides resistance through association with the antibiotic target, is the least characterized[2], and the mechanism is unique for each target. One such example is FusB-type resistance to fusidic acid (FA)[3].

FA is an antibiotic that primarily targets Gram-positive bacteria, by inhibiting bacterial protein synthesis. It was introduced in the 1960s[4] and is used topically against *Staphylococcus aureus* infections. FA binds to elongation factor G (EF-G), the five-domain (I–V) GTPase translation factor involved in tRNA translocation[5] and ribosome recycling[6], preventing its release from the ribosome[7,8]. The binding site of FA is an interdomain pocket of EF-G lined by the sarcin-ricin loop (SRL) of 23S rRNA. This pocket is only formed in the ribosome-bound EF-G after GTP hydrolysis[9] when switch I has transitioned to the GDP conformation. FA has been shown to stabilize switch II in its GTP conformation,

inhibiting the conformational changes of EF-G required for release from the ribosome[9]. There are structures of FA-locked EF-G complexes in two ribosomal states: the chimeric state, where the head of the ribosomal small subunit (SSU) is swiveled, and tRNA translocation is not completed[10], and the post-translocational state[9], where the ribosome and the tRNAs are in the classical state.

The clinically most prevalent resistance mechanism to FA is FusB-type resistance, involving a resistance protein, FusB (or the homologs FusC, FusD or FusF), that provides low-level resistance to FA (minimum inhibitory concentration of 4–16 μg/mL, compared to 0.125 μg/mL for susceptible strains)[11–13] through binding to EF-G[14]. In addition, resistance can occur through mutations in EF-G[15], with direct effects on FA binding, or affecting EF-G stability or EF-G-ribosome interactions[16]. In laboratory experiments, mutations in uL6[17], which interacts with EF-G[18] can also provide resistance.

FusB is a two-domain protein, with an N-terminal alpha-helical bundle domain (NTD), and a C-terminal treble-clef zinc finger domain

[1]Department of Cell and Molecular Biology, Uppsala University, BMC, Uppsala, Sweden. [2]Uppsala Antibiotic Center, Uppsala University, Uppsala, Sweden. [3]Proteomics Core Facility, Scilifelab and University of Gothenburg, Gothenburg, Sweden. ✉e-mail: maria.selmer@icm.uu.se

(CTD)[19]. FusB binds with high affinity to *S. aureus* EF-G[20], forming interactions with domains IV–V of EF-G[19,20]. Nuclear magnetic resonance (NMR) studies have shown that FusB binding causes increased dynamics of domain III of EF-G[21,22]. It has thus been proposed that FusB binding allosterically promotes disorder in domain III of EF-G, which causes release from the FA-inhibited ribosome[22]. FusB also increases the turnover of *S. aureus* EF-G on the ribosome in the absence of FA[23].

Here, we set out to determine the detailed structural mechanism of FusB-mediated rescue of FA inhibition. Using time-resolved cryo-EM, we solve multiple structures at different stages of rescue, including a pre-release complex that reveals how FusB releases EF-G from the ribosome and causes FA resistance. We additionally observe independent binding of FusB to the ribosomal A-site, supported by affinity measurements, but in vitro biochemistry shows no effect on aminoacyl-tRNA delivery. We describe an intricate mechanism of resistance that involves both FusB-mediated changes in the conformation of EF-G, and direct FusB-ribosome interactions.

## Results

### Capturing FusB-mediated FA rescue

To capture FusB-mediated rescue using time-resolved cryo-EM, we prepared a FA-locked complex containing 70S ribosomes, a short synthetic mRNA, *E. coli* tRNA^fMet, and *S. aureus* EF-G, in the presence of FA and GTP. Initial tests with *S. aureus* 70S showed too low particle density for characterization of lowly populated states. Since FusB does not bind to *E. coli* EF-G[3], we used the previously validated[19] heterologous *E. coli* 70S-*S. aureus* EF-G complex for most of the work, but validated the highly populated states with *S. aureus* 70S. To allow the capture of short-lived rescue complexes on the cryo-EM grid with a standard plunging setup, the FA-locked ribosome complex was first incubated on the grid, followed by addition of FusB immediately before blotting and plunge-freezing (Supplementary Fig. 1). Initial tests showed that two datasets collected on different parts of the same grid contained strikingly different populations of complexes (Supplementary Figs. 2 and 3), demonstrating that incomplete mixing of FusB with FA-inhibited 70S produced a diffusion-dependent gradient of ribosomal states. This mix-on-grid method allowed grid vitrification 6 s after FusB addition, which is thus the upper time limit for the reaction of the FA-inhibited complexes with FusB. To obtain a second time point, a dataset was collected from an identical sample conventionally mixed in a tube 25 s before plunge-freezing.

Through extensive focused 3D classification (Supplementary Figs. 4 and 5) around the A-site region using a large number of classes, we identified six different complexes relevant to FA inhibition and rescue (Fig. 1): FA-locked EF-G in chimeric state (CHI), FA-locked EF-G in post-translocational state (POST), FusB bound to EF-G in canonical POST state (FusB•EF-G•70S), FusB bound to domains IV–V of EF-G in a non-canonical ribosome-binding site (FusB•EF-G•70S*), FusB bound to the decoding center of the SSU independently of EF-G (FusB•70S:SSU), and FusB bound to the large ribosomal subunit (LSU) and the P-site tRNA independently of EF-G (FusB•70S:LSU). The global resolution of the maps ranges from 1.87 to 2.79 Å (Table 1), with local resolutions of EF-G and FusB ranging from 2.5 to 5 Å (Supplementary Fig. 6).

The complexes show different abundance at the two time points, demonstrating FusB-mediated rescue of FA inhibition (Fig. 1). The pre-rescue FA-locked EF-G complexes (CHI and POST) dramatically decrease in population between the early and late time points from 12.7% to 2.7% of the total ribosome population. The canonical rescue complex FusB•EF-G•70S is only present in the early dataset and at a relatively low proportion (1.6%), indicating that it is an early and transient complex and that most ribosomes have already been rescued at this time point. The other FusB•EF-G complex (FusB•EF-G•70S*) is present in both datasets at 1.2%. Strikingly, the FusB•70S:SSU complex constitutes 19% and 13% of the ribosomes at the two time points, showing that FusB has a strong propensity for ribosome binding also independently of EF-G.

### Locking of *S. aureus* EF-G to *E. coli* 70S by FA

The CHI and POST structures show that *S. aureus* EF-G is locked in a close to identical structure on the *E. coli* ribosome as on the *S. aureus* ribosome[18] (Supplementary Fig. 7a, b). As expected, EF-G binds in an extended conformation to the ribosome, spanning from the SSU A-site to the SRL of the LSU. FA is bound in its pocket next to GDP and a magnesium ion (Supplementary Fig. 7c, d).

### FusB•EF-G on the ribosome

In the early rescue complex, FusB•EF-G•70S, FusB is bound to the POST-state EF-G (Fig. 2a), and there is clear density showing that FA and GDP remain bound to EF-G (Supplementary Fig. 8d, h, i). In this state, FusB makes extensive contact with EF-G through both of its domains (Fig. 2b). The N-terminal domain of FusB is wedged between domains I and V of EF-G and also interacts with domains II-III of EF-G. The C-terminal domain of FusB forms an extended β-sheet with domain IV of EF-G and also interacts with domains III and V.

Alignment of the FusB•EF-G•70S and POST complex structures based on 23S rRNA shows that FusB causes a major conformational change of EF-G (Fig. 2c–g, Supplementary Fig. 8a–d, and Supplementary Movie 1). The N-terminal domain of FusB would otherwise clash with helix 3 of domain I of EF-G (Fig. 2c). Instead, it pushes domains I–III away from domains IV–V (Fig. 2d). Meanwhile, the C-terminal domain of FusB is anchored to the SSU and pulls domain IV away from the tRNA and mRNA (Fig. 2e), with a 7 Å shift of the beta-sheet of domain IV. The interactions between FusB and the ribosome mainly occur between two loops (K99-K105, K174-S177) and helices h18 and h34

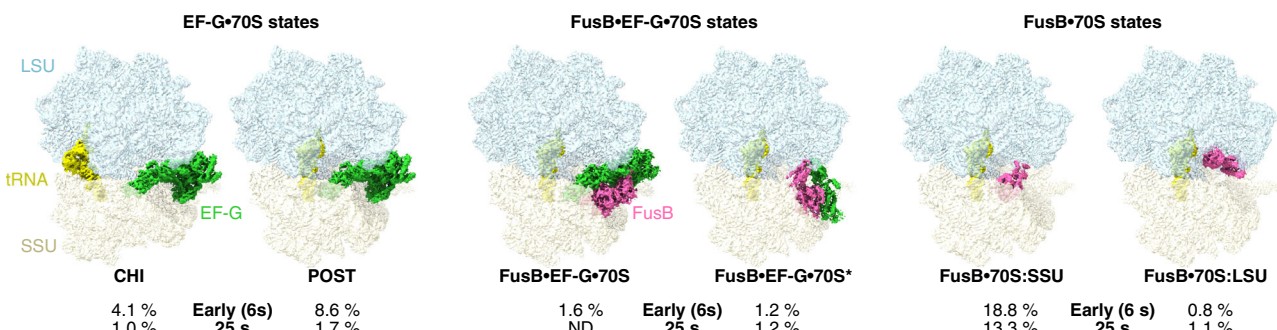

**Fig. 1 | Time-resolved cryo-EM of FusB-mediated rescue of FA inhibition.** Cryo-EM maps of six identified states are segmented 3 Å around the model and are showing the LSU (light blue), SSU (light yellow), tRNA (yellow), EF-G (green) and FusB (pink). The fraction of particles assigned to each state at each time point (or ND, not detected) is indicated. Ribosome states with empty A-site, low-occupancy EF-G or A-site tRNA correspond to the remaining particle populations (64.9% at the early time point, 6 s, and 81.7% at the 25 s time point).

**EF-G•70S states**

| CHI | POST |
|---|---|
| 4.1 % | 8.6 % |
| 1.0 % | 1.7 % |

**FusB•EF-G•70S states**

| FusB•EF-G•70S | FusB•EF-G•70S* |
|---|---|
| 1.6 % | 1.2 % |
| ND | 1.2 % |

**FusB•70S states**

| FusB•70S:SSU | FusB•70S:LSU |
|---|---|
| 18.8 % | 0.8 % |
| 13.3 % | 1.1 % |

Early (6s) / 25 s

**Table 1 | Cryo-EM refinement parameters and model validation**

| | CHI (EMDB-51350) (PDB 9GHA) | POST (EMDB-51351) (PDB 9GHB) | FusB·EF-G·70S (EMDB-51352) (PDB 9GHC) | FusB·EF-G·70S* (EMDB-51353) (PDB 9GHD) | FusB·70S:SSU (EMDB-51354) (PDB 9GHE) | FusB·70S:LSU (EMDB-51355) (PDB 9GHF) | FusB·Sa70S:SSU (EMDB-51356) (PDB 9GHG) | FusB·Sa70S:LSU (EMDB-51357) (PDB 9GHH) |
|---|---|---|---|---|---|---|---|---|
| **Data collection and processing** | | | | | | | | |
| Dataset | Early | Early | Early | 25 s | 25 s | 25 s | FusB·Sa70S | FusB·Sa70S |
| Magnification | 130,000 | 130,000 | 130,000 | 130,000 | 130,000 | 130,000 | 165,000 | 165,000 |
| Voltage (kV) | 300 | 300 | 300 | 300 | 300 | 300 | 300 | 300 |
| Electron exposure (e−/Å²) | 40 | 40 | 40 | 45.47 | 45.47 | 45.47 | 28.14 | 28.14 |
| Defocus range (μm) | −0.7 to −1.0 | −0.7 to −1.0 | −0.7 to −1.0 | −0.7 to −1.0 | −0.7 to −1.0 | −0.7 to −1.0 | −0.7 to −1.0 | −0.7 to −1.0 |
| Pixel size (Å) | 0.6480 | 0.6480 | 0.6480 | 0.6480 | 0.6478 | 0.6478 | 0.7280 | 0.7280 |
| Symmetry imposed | C1 | C1 | C1 | C1 | C1 | C1 | C1 | C1 |
| Initial particle images (no.) | 2,239,537 | 2,239,537 | 2,239,537 | 3,613,403 | 3,613,403 | 3,613,403 | 859,410 | 859,410 |
| Final particle images (no.) | 88,568 | 184,659 | 34,617 | 30,847 | 349,231 | 29,766 | 67,441 | 15,167 |
| Map resolution (Å) | 2.24 | 2.21 | 2.79 | 2.41 | 1.87 | 2.40 | 2.22 | 2.70 |
| FSC threshold | 0.143 | 0.143 | 0.143 | 0.143 | 0.143 | 0.143 | 0.143 | 0.143 |
| **Refinement** | | | | | | | | |
| Initial model used (PDB code) | 9GHE | 9GHE | 9GHE | 9GHE | 8CGK, 8CGJ, 8CF1, 7KOO, 7N2C, 4ADN | 9GHE | 8P2F, 4ADN | 9GHG |
| Model resolution (Å) | 2.2 | 2.2 | 2.8 | 2.4 | 1.9 | 2.4 | 2.2 | 2.7 |
| FSC threshold | 0.5 | 0.5 | 0.5 | 0.5 | 0.5 | 0.5 | 0.5 | 0.5 |
| Post-processing | Local filtering | Local filtering | Local filtering | Local filtering | Local filtering | Local filtering | Local filtering | Local filtering |
| Map sharpening B factor. (Å²) | −35 | −36 | −25 | −33 | −36 | −33 | −32 | −23 |
| Model composition | | | | | | | | |
| Non-hydrogen atoms | 145,564 | 145,490 | 147,172 | 143,666 | 146,257 | 141,978 | 150,404 | 144,758 |
| Protein residues | 6176 | 6188 | 6380 | 5946 | 5718 | 5718 | 5685 | 5680 |
| RNA residues | 4510 | 4499 | 4502 | 4498 | 4501 | 4501 | 4654 | 4650 |
| Waters | 0 | 0 | 0 | 0 | 4235 | 0 | 5514 | 0 |
| Magnesium atoms | 272 | 303 | 301 | 303 | 303 | 303 | 145 | 145 |
| B factors (Å²) | | | | | | | | |
| Protein | 72 | 71 | 81 | 69 | 57 | 62 | 72 | 91 |
| RNA | 57 | 54 | 62 | 51 | 45 | 50 | 65 | 79 |
| Ligand | 41 | 44 | 49 | 39 | 30 | 38 | 30 | 44 |
| Water | – | – | – | – | 24 | – | 38 | – |
| R.m.s. deviations | | | | | | | | |
| Bond lengths (Å) | 0.008 | 0.008 | 0.008 | 0.008 | 0.008 | 0.008 | 0.008 | 0.008 |
| Bond angles (°) | 1.187 | 1.183 | 1.175 | 1.191 | 1.294 | 1.195 | 1.193 | 1.198 |
| Validation | | | | | | | | |
| MolProbity score | 1.30 | 1.12 | 1.22 | 1.06 | 1.04 | 1.12 | 1.38 | 1.38 |
| Clashscore | 3.47 | 2.87 | 3.60 | 2.5 | 2.54 | 3.02 | 4.17 | 3.49 |

**Table 1 (continued) | Cryo-EM refinement parameters and model validation**

| | CHI (EMDB-51350) (PDB 9GHA) | POST (EMDB-51351) (PDB 9GHB) | FusB•EF-G•70S (EMDB-51352) (PDB 9GHC) | FusB•EF-G•70S* (EMDB-51353) (PDB 9GHD) | FusB•70S:SSU (EMDB-51354) (PDB 9GHE) | FusB•70S:LSU (EMDB-51355) (PDB 9GHF) | FusB•Sa70S:SSU (EMDB-51356) (PDB 9GHG) | FusB•Sa70S:LSU (EMDB-51357) (PDB 9GHH) |
|---|---|---|---|---|---|---|---|---|
| Poor rotamers (%) | 1.24 | 0.83 | 1.23 | 1.08 | 0.95 | 1.10 | 1.73 | 1.69 |
| Ramachandran plot | | | | | | | | |
| Favored (%) | 97.57 | 97.79 | 98.50 | 98.66 | 98.52 | 98.68 | 98.67 | 98.78 |
| Allowed (%) | 2.31 | 2.08 | 1.44 | 1.29 | 1.45 | 1.29 | 1.31 | 1.20 |
| Disallowed (%) | 0.12 | 0.13 | 0.06 | 0.05 | 0.04 | 0.04 | 0.02 | 0.02 |
| Rama-Z score | −0.54 | −0.54 | 1.19 | 0.88 | 1.13 | 1.23 | 1.07 | 0.99 |

(Supplementary Fig. 8e, f). The interactions with h34 seem to provide specificity for the POST-state ribosome since this helix is part of the SSU head that moves between the CHI and POST states.

All domains of EF-G are subject to conformational change (Fig. 2d–g and Supplementary Fig. 8a–d), while the ribosome is unaffected. This results in a 30% reduction in the contact area between EF-G and the ribosome, and broken interactions between EF-G and the P-site tRNA and mRNA (Supplementary Fig. 9 and Supplementary Table 1), causing disorder at the tip of domain IV (residues 497–504). The structural integrity of the FA-binding pocket and the density for FA (Supplementary Fig. 8g–i) show that FusB primarily promotes EF-G dissociation from the ribosome, which, in turn, leads to FA dissociation since FA has very low affinity to free EF-G[24].

### FusB binds 70S ribosomes independently of EF-G

Surprisingly, FusB also binds to two different sites of the ribosome without EF-G (Fig. 3 and Supplementary Fig. 10). In the most populated complex (FusB•70S:SSU), FusB is bound at the decoding center of the SSU of the classical-state ribosome. The C-terminal domain of FusB makes direct contact with the monitoring bases A1492 and A1493 in h44 (Fig. 3b). The side chains of the lysine-rich loop of FusB (K99-K103) interact with h31, h32, and h34 (Fig. 3c). FusB also interacts with h18, the mRNA, uS12, and the intersubunit bridge B2a.

In the FusB•70S:LSU complex, FusB contacts the LSU at H71, H89, H91, the SRL at H95, uL16, and the P-site tRNA of the classical-state ribosome (Fig. 3d–f). The region of the C-terminal domain of FusB close to the tRNA, including the lysine-rich loop (K99-K103), is more disordered than the rest of FusB.

FusB would, in both of its independent binding sites, sterically clash with EF-G in the FA-locked state, showing that FusB directly competes with EF-G to interact with the ribosome. Multiple sequence alignment (Fig. 4) shows that most of the ribosome-interacting residues are conserved also in FusC, FusD, and FusF, suggesting that these proteins would also be able to bind to the ribosome similarly. The residues involved in interactions with EF-G are similarly to a large extent conserved. Remarkably, there is a large overlap between residues in FusB that interact with the ribosome (in the FusB•70S complexes), and with EF-G (in FusB•EF-G•70S and FusB•EF-G•70S*). For example, F156 and Y187, which, if mutated to alanine, make FusB unable to bind to EF-G[20], interact directly with C1914 from 23S rRNA (Fig. 3b) and A1492 from 16S rRNA, respectively. Hence, the binding of FusB to EF-G and to its independent sites on the ribosome are mutually exclusive.

To test whether FusB binding to ribosomes could be physiologically relevant, we constructed a FusB E57C mutant carrying a single surface-exposed cysteine, and after fluorescein labeling measured the affinity of FusB to 70S ribosomes from *E. coli* and *S. aureus* using fluorescence polarization. This resulted in a $K_D$ of $320 \pm 31$ nM to the *S. aureus* 70S ribosome and $330 \pm 57$ nM to the *E. coli* 70S ribosome (Fig. 5a, b).

Next, we performed mass spectrometry-based relative quantification of FusB and EF-G in a clinical *S. aureus* isolate where expression of FusB from resistance plasmid pUB101 is controlled by translational attenuation. Under exponential growth in the presence of 2 μg/mL FA, 16-fold above the MIC for susceptible *S. aureus* strains, the molar ratio of EF-G to FusB was estimated to $2.3 \pm 0.3$ (Supplementary Table 2).

To determine if the observed binding of FusB to the ribosome would impair tRNA delivery, dipeptide, and tripeptide formation assays were performed using a single-turnover, fast-kinetics approach. Here, 70S initiation complex (IC) was pre-incubated with an excess of FusB and subsequently combined with elongation mixture (EM) in a quench-flow instrument. The presence of FusB in the IC had no effect on dipeptide formation, the rate remaining nearly identical, at $36.6 \pm 1.2$ s$^{-1}$ without FusB and $35.6 \pm 1.3$ s$^{-1}$ with FusB (Fig. 5c). This indicates that FusB binding to the ribosome does not inhibit EF-Tu-

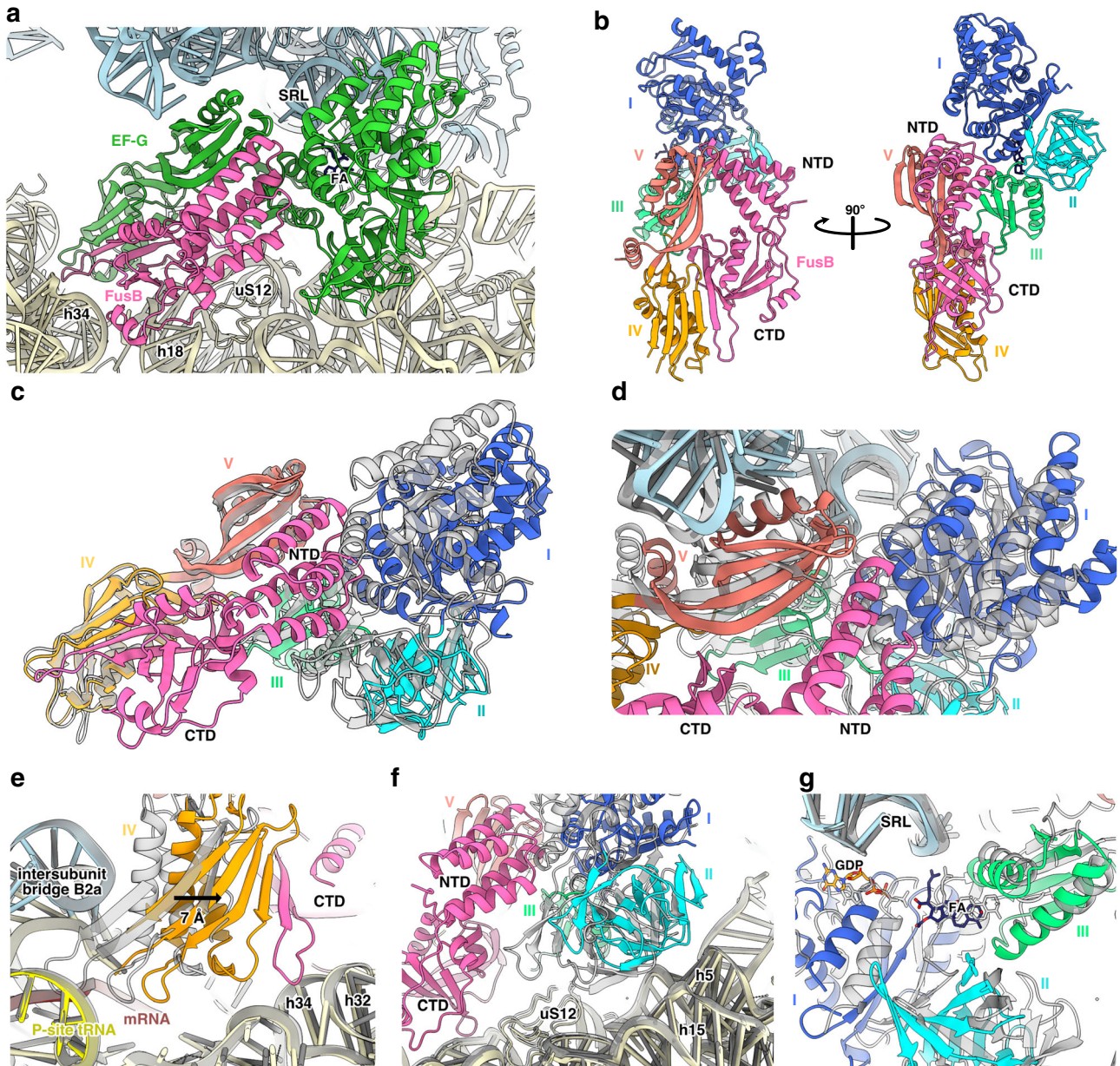

**Fig. 2 | Structure of the FusB•EF-G•70S FA rescue complex. a** FusB (pink) bound to FA-locked EF-G (green). **b** Overview of the FusB•EF-G complex, EF-G colored by domain (I, blue; II, cyan; III, light green; IV, orange; V, salmon). **c** Comparison of EF-G in FusB•EF-G•70S (colors as in (**b**)) with EF-G in the POST conformation (gray) when aligned to domains IV–V. **d**–**g** Movement of EF-G domains from the POST state (gray) to the FusB•EF-G•70S complex, with structures aligned by 23S rRNA (colors as in (**b**)); tRNA, yellow; mRNA, brown; FA, dark blue).

mediated tRNA delivery. FusB, however, did decrease the tripeptide formation rate by approximately 2.5-fold when *S. aureus* EF-G was used, from $0.54 \pm 0.01\,\text{s}^{-1}$ in the absence of FusB to $0.22 \pm 0.09\,\text{s}^{-1}$ when FusB was present (Fig. 5d). After dipeptide formation, both the independent binding sites of FusB would be sterically blocked by the A-site tRNA. Thus, inhibition of tripeptide formation is likely due to sequestering of EF-G, since FusB has high affinity to *S. aureus* EF-G in solution[23]. This explanation is further supported by a control tripeptide formation experiment with *E. coli* EF-G, to which FusB does not bind[3], showing no inhibitory effect (Fig. 5e).

## FusB binds identically to both *E. coli* and *S. aureus* ribosomes
To further validate the observed binding of FusB to the ribosome, a control cryo-EM sample was prepared with *S. aureus* 70S ribosomes, *E. coli* tRNA^fMet, a short synthetic mRNA, and FusB (in the absence of EF-G

and FA). From the cryo-EM data processing (Supplementary Fig. 11), it was observed that FusB formed the same two FusB•70S complexes with *S. aureus* 70S: FusB•Sa70S:SSU and FusB•Sa70S:LSU. In both cases, FusB and its interactions with the ribosome are close to identical to the heterologous complexes (Supplementary Fig. 12). This is not surprising, as all the interacting nucleotides and amino acid residues are conserved between *E. coli* and *S. aureus*, except for R51 (I52 in *S. aureus*) in uL16. In this dataset, FusB was found in 12.3% of the total ribosomes (10% FusB•Sa70S:SSU and 2.3% FusB•Sa70S:LSU).

## FusB•EF-G binds to the ribosome outside of the A-site
One of the identified complexes (FusB•EF-G•70S*) shows FusB bound to domains IV–V of EF-G, with no visible density for domains I–III of EF-G (Fig. 1). In this state, domain IV of EF-G does not contact the ribosome (Fig. 6a). Domain V interacts with the SRL, the stem-loops of

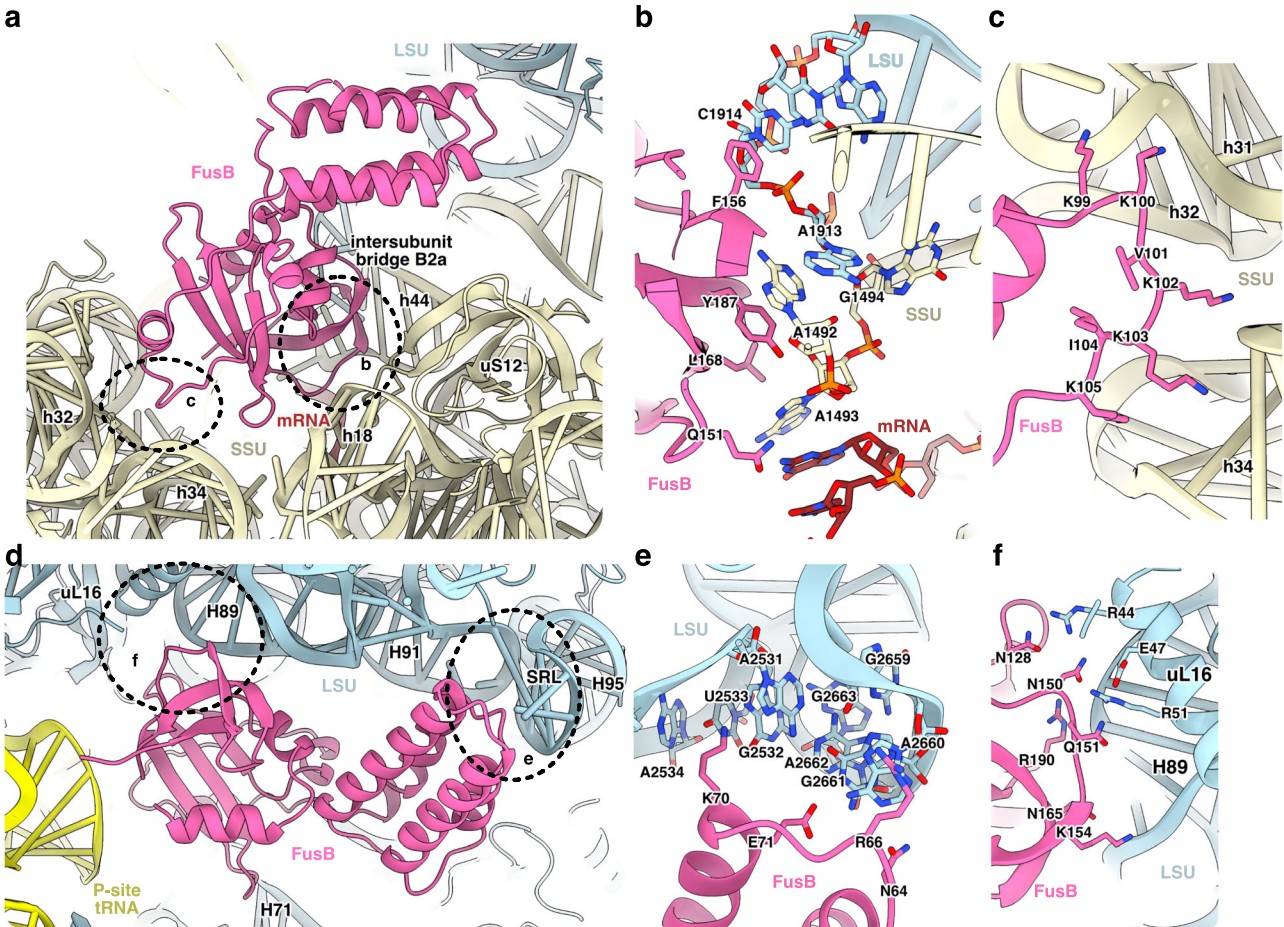

**Fig. 3 | Independent binding of FusB to the ribosome. a** SSU (light yellow) binding site of FusB (pink) in FusB•70S:SSU (LSU, light blue; mRNA, brown). **b** FusB interaction with intersubunit bridge B2a. **c** The lysine-rich loop of FusB (K99-K103) contacts h31, h32, and h34 of 16S rRNA. **d** LSU binding site of FusB (FusB•70S:LSU). **e** Interaction of FusB with H91 and the SRL of 23S rRNA. **f** FusB interaction with H89 and uL16.

H43 and H44, and uL6. FusB contacts h18 (through its lysine-rich loop, K99-K103), h34, the stem-loops of H43 and H89, and uS12. The FusB•EF-G interface is the same as in FusB•EF-G•70S (Fig. 6b), apart from interactions with the unresolved domains I–III of EF-G. If these would retain the same conformation as in FusB•EF-G•70S, they would clash with the LSU. Furthermore, the conformations of these domains in crystal structures of EF-G would also clash with FusB (Supplementary Fig. 13). This indicates that domains I–III are likely disordered relative to domains IV–V.

The orientation of this complex differs by almost 180° compared to FusB•EF-G•70S. Therefore, it is unlikely to represent a late rescue intermediate as it would require a large rotation of FusB•EF-G, while keeping domain V of EF-G between H44 and H95. Since the proportion of FusB•EF-G•70S* is the same at both time points, this more likely represents a rebinding event of FusB•EF-G.

## Discussion
In this work, we have solved the structure of a short-lived antibiotic resistance rescue complex of FusB bound to the FA-inhibited ribosome. This state could not be observed at 25 s after reaction start, which is the shortest time point we could achieve by conventional plunge-freezing. Instead, we devised the mix-on-grid method (Supplementary Fig. 1), taking advantage of the slow on-grid diffusion of FusB, which effectively created a concentration distribution, and thus a reaction time distribution of ribosomal complexes on the grid. The majority of FA-inhibited ribosomes had already been rescued, but this simple methodology in combination with advanced particle classification gave access

to high-resolution information of an otherwise non-observable state in this process. We envision that this procedure will also be applicable and provide insights into other biological processes since it offers a readily accessible approach in the absence of advanced instrumentation for time-resolved grid preparation.

Using this methodology, we obtained structures of six different complexes related to FusB-mediated rescue of FA inhibition, using a previously validated heterologous system composed of *E. coli* 70S ribosomes and *S. aureus* FusB and EF-G[19,23]. Based on these, we propose a detailed mechanism of FusB-type resistance (Fig. 7). FA locks EF-G to the ribosome in two possibly interchanging states (CHI and POST structures). FusB binds to EF-G in the POST state (FusB•EF-G•70S). Here, FusB induces a major conformational rearrangement of EF-G by wedging its N-terminal domain between domains I and V and by interacting with both the SSU and domain IV of EF-G. This leads to loss of EF-G contacts with the ribosome, tRNA, and mRNA, causing release of the FusB•EF-G complex from the ribosome, followed by dissociation of FA from EF-G to allow subsequent binding of ternary complex. In comparison with effects of different FA resistance mutations in EF-G (FusA), which interfere with direct FA interactions, EF-G-ribosome interactions, structural stability of EF-G and interdomain contacts in EF-G[16], FusB simultaneously alters of all these properties apart from the direct FA-EF-G interactions. FusB is also observed to bind to sites in either subunit in the empty A-site of the classical-state 70S ribosome, which would both sterically clash with the FA-locked conformation of EF-G. These binding sites were validated in structures of the *S. aureus* ribosome.

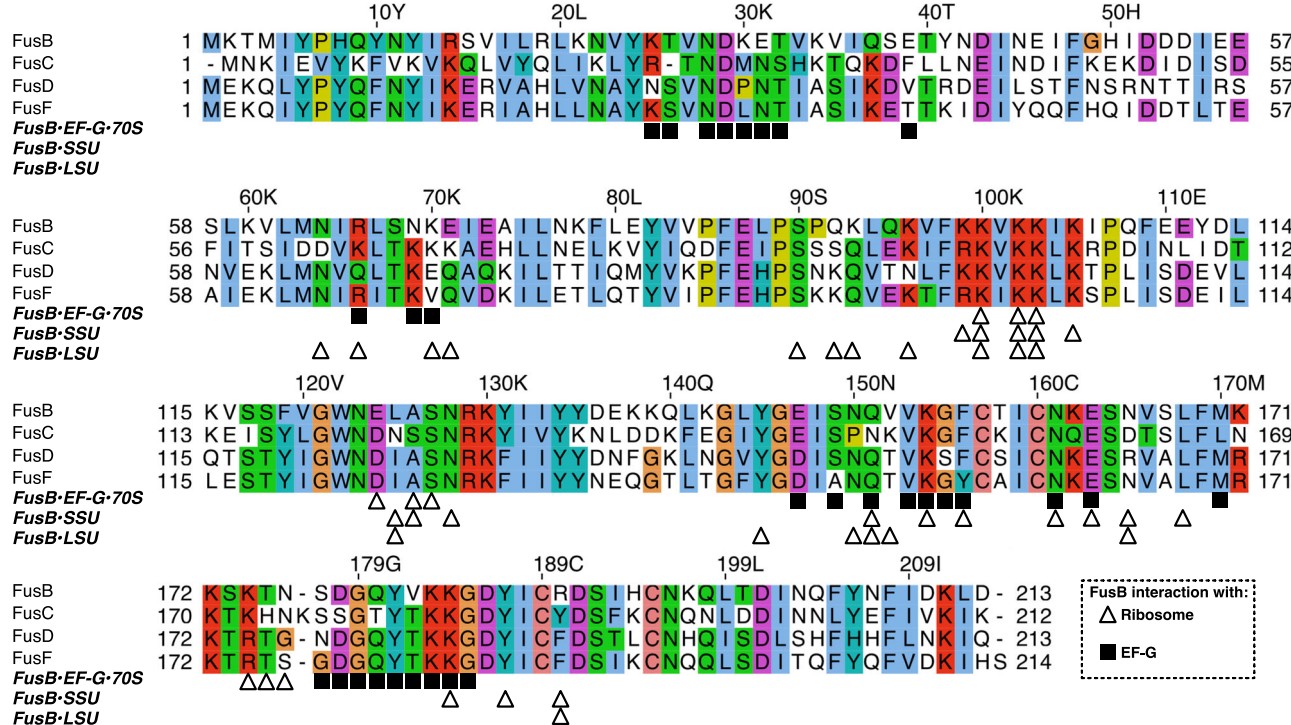

**Fig. 4 | Multiple sequence alignment of FusB and a selection of homologs.** Alignment between FusB (Uniprot Q8GNY5), FusC (WP_001033157), FusD (WP_011303797), and FusF (BAQ33930) using the ClustalX coloring scheme.

Residues involved in interactions with EF-G (squares) or the ribosome (triangles) in each of the structures are marked below the alignment.

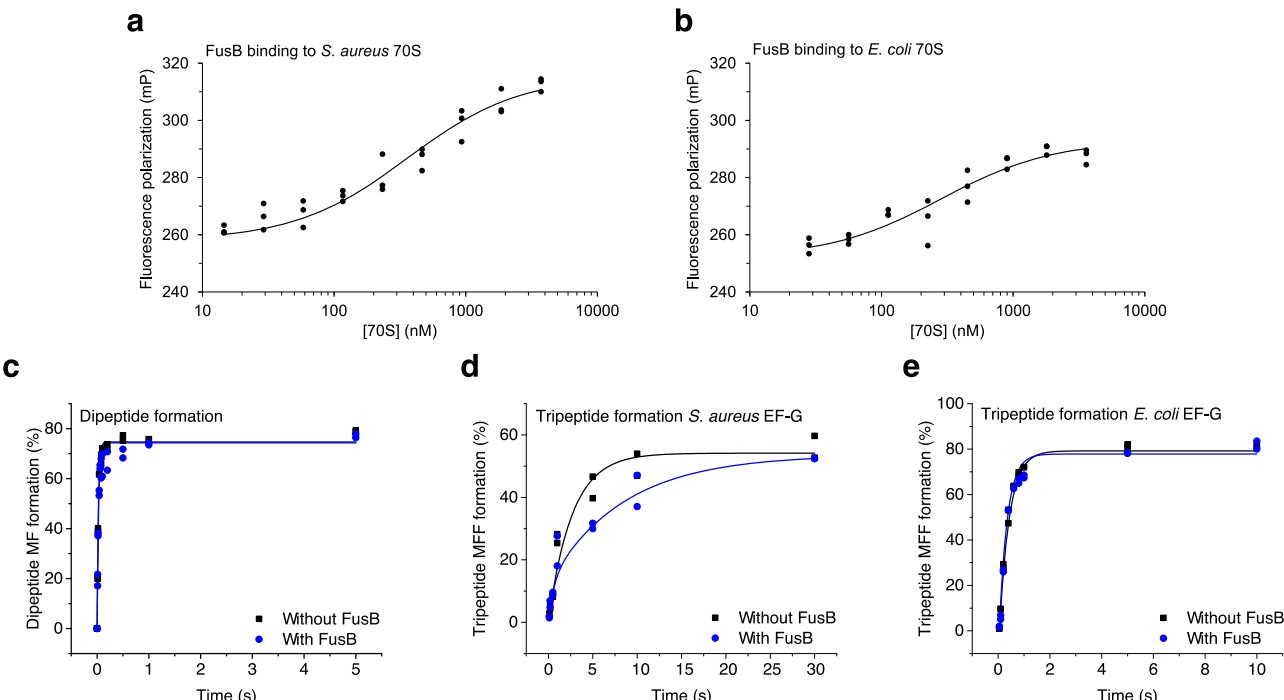

**Fig. 5 | Measurements of FusB-ribosome affinity and effects on di- and tripeptide formation. a, b** Fluorescence polarization binding experiments measuring affinities of fluorescein-labeled FusB to 70S ribosomes from *S. aureus* (**a**) and *E. coli* (**b**) in $n = 3$ independent measurements (technical replicates in Supplementary Fig. 16). **c–e** Di- and tripeptide formation experiments in the absence (black) and presence (blue) of FusB in $n = 2$ independent experiments. **c** Comparison of single-turnover dipeptide formation with *S. aureus* EF-G. **d** Comparison of single-turnover tripeptide formation with *S. aureus* EF-G. **e** Comparison of single-turnover tripeptide formation with *E. coli* EF-G. Source data are provided as a Source Data file.

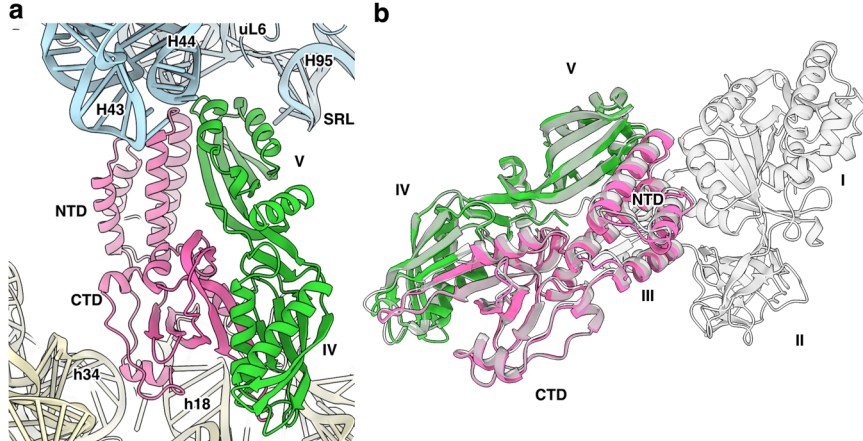

**Fig. 6 | Non-canonical ribosome binding of FusB•EF-G in FusB•EF-G•70S*.**
**a** Binding site of FusB (pink) and EF-G (green) between the LSU (light blue) and the SSU (light yellow). **b** Comparison of the FusB•EF-G complex in FusB•EF-G•70S* (pink and green) and FusB•EF-G•70S (gray). The structures are aligned to EF-G domains IV–V. The RMSD is 0.589 Å over 204 $C_\alpha$ atoms for EF-G, and 0.582 Å over 212 $C_\alpha$ atoms for FusB.

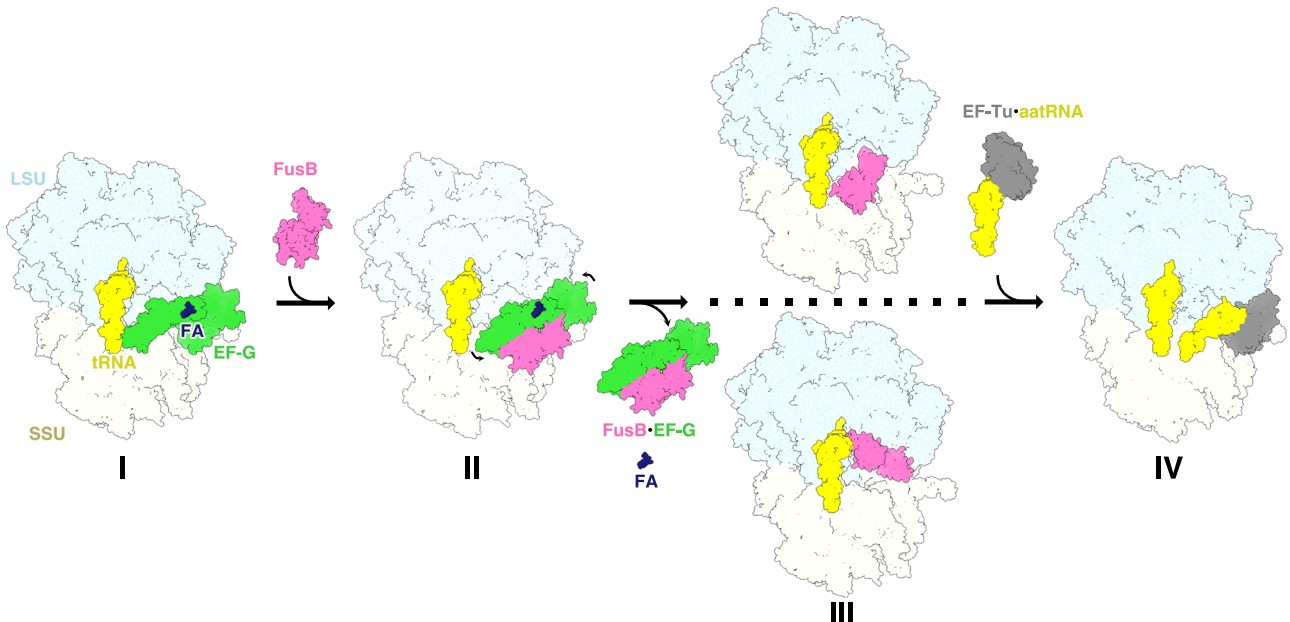

**Fig. 7 | Proposed mechanism of FusB-mediated resistance.** FusB (pink) binds to the FA-locked EF-G (green) (I), inducing conformational changes and breaking EF-G contacts to the ribosome (LSU, light blue; SSU, light yellow; tRNA, yellow), mRNA and tRNA (II), leading to EF-G release followed by FA dissociation. FusB shows binding to two parts of the ribosomal A-site in absence of EF-G (III), but does not inhibit binding of ternary complex (gray and yellow) (IV).

During the canonical translocation cycle of EF-G, GTP hydrolysis is followed by conformational changes of the switch regions of domain I, inducing interdomain conformational changes required for dissociation from the ribosome[25]. Time-resolved cryo-EM studies of translocation indicate that domains I–III of EF-G dissociate first from the ribosome, while domain IV retains its normal position until release[26]. Binding of FusB was previously proposed to cause release of the FA-inhibited state "by allosteric effects on the dynamics of the drug target"[22]. Here, we have instead shown that EF-G is released in the presence of FA through a conformational pathway involving loss of contact with the ribosome, tRNA, and mRNA, particularly for domain IV of EF-G. Our structures suggest that the increased dynamics of domain III observed in the free FusB•EF-G complex[22] are a consequence of absent contacts with the ribosome in combination with steric clashes of FusB with stabilizing interdomain contacts observed in crystal structures of EF-G (Supplementary Fig. 13). Domain III is the most dynamic of the domains in EF-G, and displays disorder in many EF-G structures[16,27]. The steric block by FusB seems to uncouple domains I–II from domains IV–V of EF-G, explaining why only domains IV and V of EF-G are ordered relative to the ribosome in the non-canonical FusB•EF-G•70S* complex.

Our structures of the FusB-EF-G complex on the ribosome (FusB•EF-G•70S and FusB•EF-G•70S*) show large discrepancies with a published NMR-based model of FusB in complex with domains III-V of EF-G[21] (Supplementary Fig. 14), for example in the interaction between the C-terminal domain of FusB and domain IV of EF-G. We find it likely that the interface between FusB and domains IV–V of EF-G observed in our ribosome complex structures will be retained in the free FusB•EF-G complex. In solution, the $K_D$ between FusB and EF-G was previously determined to ~63 nM[23]. Based on the observed interactions between FusB and EF-G in FusB•EF-G•70S, we expect the affinity between the two proteins to be similar or higher on the ribosome than in solution[23]. Similar binding of FusB to EF-G on and off the ribosome in the absence of FA could explain the observation that it causes increased turnover of

EF-G[23]. Because of the regulation of FusB expression by translational attenuation[3], this is unlikely to happen in vivo.

Remarkably, the FusB•70S complexes are at both time points more abundant than EF-G•70S complexes in our sample. Even though FusB would directly clash with an A-site tRNA, we observe no inhibition of dipeptide formation (Fig. 5c) by 10 μM FusB (10:1 to 70S and 1:1 to EF-Tu). However, under these conditions, FusB inhibits EF-G-catalyzed translocation, detected as a two-fold rate reduction of tripeptide formation at a FusB:EF-G ratio of 5:1. This is likely due to sequestering of free EF-G by FusB[3]. The complex formation will reduce the concentration of free, active, EF-G, but in addition we cannot exclude that that the FusB•EF-G complex may show some degree of translocation activity.

Production of FusB from the clinically isolated pUB101 plasmid is regulated by translational attenuation[3]. FA acts as inducer of FusB expression by stalling ribosomes on a leader sequence. This leads to changes in the mRNA structure that make the Shine-Dalgarno sequence of the FusB gene available for ribosome binding[3]. Thus, a requirement for carefully controlled concentration of FusB to provide rescue and not inhibition at physiological conditions is fulfilled. In conjunction with our structural and biochemical results, it prompted us to estimate the concentration of FusB in a clinical strain under inducing conditions (2 μg/ml FA). Under exponential growth, the concentration of EF-G in the *S. aureus* cell has been estimated to 10 μM[28]. With a molar ratio of EF-G to FusB of 2.3 ± 0.3, this indicates a FusB concentration in the low micromolar range. Thus, both FusB and the 70S ribosome are present at concentrations above the measured ~320 nM affinity of FusB to ribosomes from *E. coli* or *S. aureus*. Our data thus suggest that the direct binding of FusB to the ribosome can also occur in the cell, but is limited by the fraction of ribosomes with empty A-site as well as by the fivefold higher affinity binding to the abundant EF-G. Our analysis does not clarify to what extent FusB will bind to EF-G or ribosomes in the cell, but the measured concentrations and affinities suggest that EF-G binding will be preferred.

The direct ribosome binding of FusB could potentially have a function unrelated to FA resistance. Housekeeping proteins relating to the translation machinery provide a reservoir with potential for evolution of antibiotic resistance proteins providing target modification as well as target protection[2]. Homologs of FusB encoded in the chromosome of some Gram-positive bacteria (e.g., *Listeria, Enterococci*) that are unlikely to have encountered the antibiotic[3] may represent such housekeeping proteins that by chance provide FA resistance[3].

In contrast to most other target protection mechanisms of resistance, such as Tet(M)- and ABC-F-mediated resistance[2], FusB does not interact directly with the antibiotic, and its action does not directly induce the release of the antibiotic. Instead, the target protein (EF-G) is released from the ribosome, which in turn leads to the dissociation of the antibiotic, as FA does not bind to EF-G in solution[24]. Unfortunately, this makes the development of FA analogs or derivatives that overcome FusB-type resistance challenging, but future approaches could aim for FA derivatives with increased ribosome interactions or inhibitors of FusB-EF-G binding.

In this study, we have elucidated the structural mechanism of FusB-mediated rescue of FA inhibition, but some details still remain unclear. In a complex equilibrium, FusB can bind to EF-G on and off the ribosome and to two ribosomal binding sites. Future studies will clarify this interplay in vivo, as well as the function of ribosome binding of FusB or its homologs.

## Methods

### Cloning, overexpression, and purification of *S. aureus* FusB and EF-G

*S. aureus* EF-G was purified as published[18] by immobilized metal chromatography and size exclusion chromatography. The final protein sample was stored in 50 mM Tris-HCl pH 7.5, 300 mM NaCl, 5 mM β-mercaptoethanol at −70 °C. This construct includes a 6xHis tag with a TEV protease cleavage site and a linker to the EF-G sequence. After TEV cleavage, only Ser-Leu remains at the N-terminus of EF-G.

The *S. aureus fusB* gene was cloned into pEXP5-NT-TOPO (Thermo Fisher Scientific, Waltham, MA, USA) following the manufacturer's instructions using primers 5′-ATGAAGACAATGATTTATCCTCAC-3′ and 5′-CACAAACATAGTTAATTCCTTAATCTAG-3′ for PCR amplification of the insert. The sequence of pEXP5-NT_FusB was confirmed by Sanger sequencing. This construct encodes a 6xHis tag with a TEV protease cleavage site and a linker to the FusB sequence. After TEV cleavage, only Ser-Leu remains at the N-terminus. FusB mutant E57C was constructed through QuikChange (Agilent Technologies, Inc., Santa Clara, CA, United States) mutagenesis following the manufacturer's instructions using the following primers: 5′- GGTCATATAGATGACGA-TATTGAATGCTCTTTAAAAGTATTAATGAACATC-3′, and 5′- GATGTT CATTAATACTTTTAAAGAGCATTCAATATCGTCATCTATATGACC-3′.

pEXP5-NT_FusB or pEXP5-NT_FusB_E57C was transformed into *Escherichia coli* BL21 (DE3). An overnight culture was inoculated 1:100 into a 2.8 L baffled flask with 800 mL of LB and 100 μg/mL of ampicillin. Protein expression was induced at an $OD_{600}$ of 0.5−0.6 with 1 mM isopropyl-β-thiogalactopyranoside and the culture was incubated for 16−18 h at 30 °C. The cells were harvested at 7400×*g* for 30 min in a JLA 9.1000 rotor (Beckman Coulter, Brea, California, USA), washed with 150 mM NaCl, and stored at −20 °C.

The cells were resuspended in lysis buffer (50 mM Tris-HCl pH 7.5, 300 mM NaCl, 20 mM imidazole) with 0.1% (v/v) Triton X-100, one EDTA-free mini-Complete protease inhibitor tablet (Roche, Basel, Switzerland) and DNase I. Then, the resuspended cells were lysed in a flow cell disruptor (Constant Systems Ltd., Daventry, UK). The lysate was centrifuged at 27,000×*g* for 45 min in an SS-34 rotor (Thermo Fisher Scientific).

The supernatant was filtered by 0.45 μm with a polyethersulfone syringe filter (Sarstedt AG & Co, Nümbrecht, Germany) and incubated with 1 mL of Ni Sepharose Fast Flow (Cytiva, Uppsala, Sweden) equilibrated with lysis buffer. The column was washed with 25 column volumes of 600 mM NaCl in lysis buffer at pH 7.8, and the protein was eluted with 400 mM imidazole in lysis buffer at pH 7.8. The eluate was exchanged into GF buffer (20 mM Tris-HCl pH 7.8, 300 mM NaCl, 5 mM β-mercaptoethanol) using a PD-10 column (Cytiva, Uppsala, Sweden). The 6x-histidine-tag was removed by cleavage with 1:25 molar ratio of TEV protease at 8 °C for 16−18 h followed by reverse immobilized affinity chromatography on 1 mL of Ni Sepharose Fast Flow. FusB was purified further by gel filtration on a Hiload 16/60 Superdex-75 column (GE Healthcare, Uppsala, Sweden) equilibrated with GF buffer. The peak fractions were concentrated to 5−10 mg/mL with a 10-kDa cutoff Vivaspin Turbo 15 (Sartorius AG, Göttingen, Germany), frozen in liquid nitrogen, and stored at −70 °C.

### Preparation of components for cryo-EM

*E. coli* 70S ribosomes were purified from *E. coli* MRE600 cells according to previous protocols[29]. *S. aureus* NCTC 8325-4 ribosomes were purified as published[18]. In short, cells were harvested at $OD_{600}$ of 1.0, lysed with a French press (*E. coli*) or lysostaphin (*S. aureus*), and pelleted. The supernatant was centrifuged through a 1.1 M sucrose cushion twice. The resulting crude ribosome pellet was resuspended, and 70S ribosomes were isolated through rate-zonal centrifugation in a preformed 10−40% sucrose gradient. The 70S ribosomes were pooled and pelleted. The final 70S sample was stored in HEPES polymix buffer (5 mM HEPES pH 7.5, 5 mM $NH_4Cl$, 5 mM Mg(OAc)$_2$, 100 mM KCl, 0.5 mM CaCl$_2$, 8 mM putrescine, 1 mM spermidine, and 1 mM dithioerythritol) at −70 °C. mRNA Z4AUGGCA (5′-GGCAAGGAG-GUAAAAAUGGCAAAA-3′) was produced by chemical synthesis (Dharmacon, Lafayette, United States) and *E. coli* tRNA^fMet was overexpressed and purified as published[30].

## EF-G•Ec70S complex preparation for grid vitrification

The sample was prepared by mixing 0.75 μM (final concentrations) 70S *E. coli* ribosomes in HEPES polymix buffer (with 20 mM HEPES pH 7.5 and 5 mM BME as reducing agent) with 3 μM mRNA Z4AUGGCA and incubated for 10 min at 37 °C. Then, 3 μM *E. coli* tRNA^fMet was added and the mix was incubated for 10 min at 37 °C. Next, 7.5 μM *S. aureus* EF-G, 400 μM FA (Sigma-Aldrich, Merck, Darmstadt, Germany), and 1.5 mM GTP were added, followed by incubation for 10 min at 37 °C, then kept on ice until plunge-freezing.

## FusB•Sa70S complex preparation for grid vitrification

The sample was prepared by mixing 0.5 μM (final concentrations) 70S *S. aureus* ribosomes in HEPES polymix buffer with 5 μM mRNA Z4AUGGCA and incubated for 10 min at 37 °C. Then, 5 μM *E. coli* tRNA^fMet was added and the mix was incubated for 10 min at 37 °C. Less than 30 s before plunge-freezing, 15 μM FusB was added.

## Cryo-EM grid preparation

All grids were prepared in a similar way with minor differences.

**Early sample.** A QuantiFoil 200-mesh R 2/1 copper grid with 2 nm continuous carbon (QuantiFoil Micro Tools GmbH, Großlöbichau, Germany) was glow-discharged 15 s at 20 mA and 0.39 mBar using EasiGlow (Ted Pella, Inc., Redding, CA, USA). In total, 3 μL of EF-G•Ec70S was incubated on the grid for 30 s at 4 °C and 95% humidity. Then, 1.2 μL of FusB 18.5 μM in HEPES polymix buffer with 400 μM FA was added into the drop, blotted for 3 s, and plunge-frozen into liquid ethane in a Vitrobot Mark IV (Thermo Fisher Scientific) at 4 °C and 95% humidity. The total time from adding FusB and the grid touching the ethane was 6 s.

**25 s sample.** In total, 2 μL of 18.5 μM FusB was mixed with 5 μL of EF-G•Ec70S on ice and directly added to a glow-discharged (same as the early sample) QuantiFoil 200-mesh R 2/1 copper grid with 2 nm continuous carbon, blotted for 4 s and plunge-frozen in a Vitrobot Mark IV. The total time between adding FusB and the grid touching the ethane was 25 s.

**FusB•Sa70S sample.** A QuantiFoil 300-mesh R 2/2 copper grid with 2 nm continuous carbon was glow-discharged for 30 s. Then, 3 μL of FusB•Sa70S sample was added to the grid, incubated for 10 s, blotted for 3 s, and plunge-frozen in a Vitrobot Mark IV.

**Preliminary dataset.** A QuantiFoil 300-mesh R 2/2 copper grid with 2 nm continuous carbon was glow-discharged for 30 s. 2.5 μL of EF-G•Ec70S (0.25 μM 70S ribosomes, 1 μM mRNA, 1 μM tRNA^fMet, 2.5 μM EF-G, 400 μM FA, and 0.5 mM GTP) was incubated on the grid for 30 s. Then, 1 μL of FusB 12.5 μM in HEPES polymix buffer with 400 μM FA was mixed into the drop by pipetting, blotted for 4 s, and plunge-frozen in a Vitrobot Mark IV at 4 °C and 95% humidity. The total time from adding FusB and the grid touching the ethane was 10 s.

## Cryo-EM data collection

All the grids were screened on a Glacios TEM operated at 200 kV equipped with a Falcon-III direct electron detector (Thermo Fisher Scientific, Waltham, MA, USA).

**Early dataset.** Collected on a Titan Krios G3i (Thermo Fisher Scientific, Waltham, MA, USA) operated at 300 kV and equipped with a K3 BioQuantum direct electron detector (Gatan, Inc, AMETEK, Berwyn, PA, USA) and energy filter using 20 eV slit. The data were acquired at ×130,000 nominal magnification with a calibrated pixel size of 0.648 Å. A total of 24,479 movies were collected in 40 frames with a total dose of 40 e⁻/Å2 (16.8 e⁻/pixel/s) over 1 s with a set defocus between −0.7 and −1.0 μm.

**25 s dataset.** Collected on a Titan Krios operated at 300 kV and equipped with a K3 BioQuantum direct electron detector and energy filter using 20 eV slit. The data were acquired at ×130,000 nominal magnification with a calibrated pixel size of 0.6478 Å. A total of 32,937 were collected in 45 frames with a total dose of 45.47 e⁻/Å2 (15.9 e⁻/pixel/s) over 1.2 s with a set defocus between −0.7 and −1.0 μm.

**FusB•Sa70S dataset.** Collected on a Titan Krios G2 operated at 300 kV and equipped with a Falcon-4i direct electron detector and Selectris energy filter (Thermo Fisher Scientific, Waltham, MA, USA) using 10 eV slit. The data were acquired at ×165,000 nominal magnification with a calibrated pixel size of 0.728 Å. A total of 10,065 EER formatted exposures were collected in 657 raw frames with a total dose of 28.14 e⁻/Å2 (14.06 e⁻/pixel/s) over 2.14 s with a set defocus between −0.7 and −1.3 μm.

**Preliminary dataset 1.** Collected on a Glacios TEM operated at 200 kV equipped with a Falcon-III direct electron detector. The data were acquired at ×150,000 nominal magnification with a pixel size of 0.952 Å. A total of 3490 movies were collected in 30 frames with a total dose of 30.97 e⁻/Å2 (1.01 e⁻/pixel/s) over 27.79 s with a set defocus between −0.6 and −1.4 μm.

**Preliminary dataset 2.** Collected on a Titan Krios G2 operated at 300 kV and equipped with a K3 BioQuantum direct electron detector and energy filter using 20 eV slit. The data were acquired at ×105,000 nominal magnification with a calibrated pixel size of 0.8240 Å. A total of 8533 movies were collected in 30 frames with a total dose of 28.52 e⁻/Å2 (16.1 e⁻/pixel/s) over 1 s with a set defocus between −0.8 and −1.4 μm.

## Cryo-EM data processing

All processing was done in cryoSPARC v4.4.1[31] using a similar workflow. All the final maps were post-processed by global B-factor sharpening based on linear fit to the Guinier plot and locally low-pass filtered based on local resolution estimation (cryoSPARC local filtering). The global resolution of maps was estimated within an auto-tightened mask created by cryoSPARC. For all maps used for model refinement, the resolution within a mask generated from the model was also calculated and used as the reported resolution. These masks were created by calculating a synthetic map from the model using the molmap command in ChimeraX at 20 Å, thresholded at a level where the map covers the whole model, and then adding a 16-pixel soft-edge.

**Early dataset.** The movies were motion-corrected using Patch Motion Correction, the CTFs were estimated using Patch CTF Estimation. Blob picker was used to produce an initial 3D reconstruction and produce templates for template picking. After template picking a particle stack was created by extracting a 600-pixel box around each particle. The 2D classification was used to remove obvious picking artifacts (e.g. carbon edges), followed by a homogeneous refinement performed simultaneously with higher-order CTF estimation and per-particle defocus refinement. The resulting reconstruction was used for referenced-based motion correction, and a new homogeneous refinement was performed, followed by higher-order CTF estimation and per-particle defocus refinement, resulting in a final consensus reconstruction. The 70S ribosome was subtracted using a mask that covered the ribosome with an 18-pixel soft-edge. The resulting subtracted particles were then down-sampled to 128 pixels and a focused 3D classification was performed using a mask around the ribosomal A-site that covers EF-G and the expected FusB region with an 18-pixel soft-edge. The 3D classification was performed with 120 classes, while keeping the input per-particle scales. The different resulting classes were homogeneously refined using the original full-sized non-subtracted particles. The particles in the classes containing the EF-G•70S complex were down-

sampled to 128 pixels and a focused classification inside a SSU-mask was performed with 20 classes, while keeping the input per-particle scales. Homogeneous refinement using the original particles resulted in the final CHI and POST structures. A schematic of the processing workflow is shown in Supplementary Fig. 4.

**25 s dataset.** Processing was done in a similar way as for the early dataset. The 2D classification was followed by ab initio and heterogeneous refinement with five classes to remove additional non-ribosomal particles. Focused 3D classification with an A-site mask was performed with 100 classes and using a 1% convergence criterion. The focused classification over SSU was performed with ten classes. A schematic of the processing workflow is shown in Supplementary Fig. 5.

**FusB•Sa70S dataset.** The EER movies were motion-corrected in 41 frames using Patch Motion Correction and the CTF parameters were estimated using CTFFIND4. After that, processing mostly followed the 25 s dataset. Ab initio and heterogeneous refinement to remove non-ribosomal particles were done with 4 classes. Non-uniform refinement with higher-order CTF estimation and per-particle defocus refinement was used to obtain the reference for referenced-based motion correction and then again to calculate the final consensus reconstruction. For subtracting the 70S ribosome, a mask with a 12-pixel soft-edge was used. The focused 3D classification at the A-site was done using a spherical 50 Å mask around the ribosomal A-site with a 12-pixel soft-edge and 40 classes. A schematic of the processing workflow is shown in Supplementary Fig. 11.

**Preliminary dataset 1.** Processing was done in a similar way as the early dataset. After movie preprocessing and blob picking, particles were extracted with 416-pixel box size binned to 128 pixels. Ab initio reconstruction followed by heterogeneous refinement was performed with ten classes. The ribosomal particles were re-extracted at full resolution. The consensus reconstruction was used for signal subtraction using a mask that covered the 70S ribosome with a 12-pixel soft-edge. Focused 3D classification was performed using a mask around the ribosomal A-site that covers EF-G and the expected FusB region with a 12-pixel soft-edge with 60 classes. Focused classification over SSU was performed with five classes. A schematic of the processing workflow is shown in Supplementary Fig. 2.

**Preliminary dataset 2.** Processing was done in a similar way as the early dataset. After template picking, particles were extracted with a 512-pixel box size. For focused 3D classification around the ribosomal A-site the 70S ribosome was subtracted using a mask with a 12-pixel soft-edge and using 100 classes. Focused classification over the SSU was performed with 20 classes. A schematic of the processing workflow is shown in Supplementary Fig. 3.

Local resolution estimation, Fourier shell correlation and angular distribution of the particles for the modeled structures are shown in Supplementary Fig. 15.

### Model building

All models were built by first rigid-body fitting and refining each chain in the starting model using Coot[32] against the post-processed map followed by final refinement using Servalcat[33]. Afterward, the structures were revised manually in areas relevant to this study and validation issues were corrected. Validation was performed using Phenix[34] and the MolProbity web server[35]. Refinement and validation information is found in Table 1.

**FusB•Sa70S:SSU.** The starting model was the 2.4 Å *S. aureus* FA-CP-locked structure (PDBID: 8P2F)[18]. For FusB, chain A from the crystal structure of FusB (4ADN)[19] was used, and for protein bS21 an

AlphaFold2[36] prediction using ColabFold[37] was used. Waters were added using Coot prior to the final refinement with Servalcat. The map used was from the FusB•*Sa*70S dataset.

**FusB•Sa70S:LSU.** The starting model was the refined FusB•Sa70S:SSU model. The map used was from the FusB•*Sa*70S dataset.

**FusB•70S:SSU.** The staring model consisted of PDBIDs 8CGK, 8CGJ, and 8CF1[38]. H43-44 were added from the *E. coli* FA-locked complex (PDBID 7N2C)[39], FusB, tRNA and mRNA from FusB•Sa70S:SSU and the remaining missing fragments from PDBID 7K00[40]. The map used was from the 25 s dataset.

**FusB•70S:LSU.** The starting model was the refined FusB•70S:SSU model. The map used was from the 25 s dataset.

**FusB•EF-G•70S and FusB•EF-G•70S\*.** The starting model was the refined FusB•70S:SSU model except for FusB, and FusB•EF-G from an AlphaFold2 prediction using ColabFold. The map used was from the early dataset for FusB•EF-G•70S, and the 25 s dataset for FusB•EF-G•70S\*.

**CHI and POST.** The starting model was the refined FusB•70S:SSU model. EF-G, and tRNA came from the 2.5 Å *S. aureus* FA-locked structure (PDBID: 8P2H)[18]. The maps were from the early dataset.

### Fluorescence polarization

FusB mutant E57C was labeled using Fluorescein-5-maleimide (Thermo Fisher Scientific, Waltham, MA, USA), following the manufacturer's instructions. The fluorescence polarization measurements were done at 25 °C in 384-well plates, OptiPlate black (PerkinElmer, Shelton, CT, USA), using 10 μL total volume and measured in a CLARIOstar Plus (BMG LABTECH, Ortenberg, Germany). The gain was calibrated before each run using 10 nM Fluorescein-5-maleimide with a target mP value of 35. The assays were performed with 20 nM labeled FusB and increasing concentrations of *E. coli* or *S. aureus* 70S ribosomes in triplicate wells. The sample was measured with 500 flashes over 5 cycles using an excitation filter of 482 nm, emission filter of 530 nm, and dichroic filter of 504 nm. The mP values over the five cycles were stable, and thus averaged. The data were analyzed by fitting the theoretical fluorescence polarization values derived from the equilibrium constant definitions as described before[41]. The mP value of the free labeled protein was calculated by averaging over the triplicate wells, and the fit was done by minimizing the $\chi^2$ of all the triplicate data points while varying the $K_D$ and the mP value at saturation using the solver function in Excel (Microsoft Corporation, Redmond, Washington, USA). Each assay was repeated with a technical replicate (Supplementary Fig. 16), reporting the average and the range as an estimation of the error. A singular outlier value was removed from the fitting of the *S. aureus* 70S technical replicate due to having negative fluorescence values.

### Mass spectrometry

**Sample preparation.** An overnight culture of *S. aureus* strain AH027 (clinical strain containing the pUB101 plasmid encoding for FusB) was inoculated 1:100 into tryptic soy broth (Merck, Darmstadt, Germany) supplemented with 2 μg/mL of fusidic acid and grown to -0.3 $OD_{600}$. The cells were harvested by centrifugation at 4000×*g* and washed three times with PBS. The cell pellet was stored at −20 °C until the analysis. Then, proteins were extracted in 2% sodium dodecyl sulfate (SDS) and protein concentrations were determined using Bicinchoninic acid assay. Cell sample were processed using a modified SP3 method[42]. The sample was reduced in 10 mM dithiothreitol at 56 °C for 30 min and alkylated in 20 mM iodoacetamide at room temperature for 30 min. Washed hydrophobic and hydrophilic Sera-Mag™

SpeedBeads (Carboxylate-Modified, Cytiva) were added to the sample with a bead-to-protein ratio of 10:1. Proteins were precipitated on the beads by acetonitrile (final concentration 70%), washed with 70% ethanol and 100% acetonitrile and dried at room temperature. Beads were resuspended in 50 μL 50 mM HEPES buffer pH 8 (Thermo Scientific) and proteins were digested with Trypsin/Lys-C mix [1:25, Promega], at 37 °C, for 2 h and trypsin [1:50, Thermo Scientific], at 37 °C, overnight. The peptide supernatants were collected, the magnetic beads were washed with 50 μL 50 mM HEPES buffer and combined with the supernatants. Pure recombinant FusB and EF-G were used for relative quantification. Each of the proteins EFG (60 μg) and FusB (80 μg) were digested with trypsin (Thermo Scientific), ratio 1:20 in 100 mM TEAB, total volume 50 μL, at 37 °C during 4 h.

**Mass spectrometry isobaric mass tagging.** Each peptide mix standard and the *S. aureus* cell sample were labeled using TMTpro 16-plex isobaric mass tagging reagents (Thermo Scientific). EF-G peptides (30 μg) were labeled with 133 C, FusB peptides (60 μg) were labeled with 126 and *S. aureus* lysate peptides (30 μg) were labeled with 135 N according to the manufacturer´s instructions. The final labeled solution contained 40% acetonitrile, and the final concentrations were 0.205 μg/μL EF-G, 0.312 μg/μL FusB and 0.205 μg/μL *S. aureus* peptides. The labeled samples were pooled into different mixtures of TMTpro 3-plex sets. In the first quantification test, 20 μL *S. aureus* sample peptides were dried and reconstituted in 44 μL 3% acetonitrile, 0.1% trifluoroacetic for the TMT mix. The EFG solution was diluted (2–100 μL), and 2 μL was added to the mix. The FusB solution was diluted (4–100 μL), and 4 μL added to the mix to reach a total volume of 50 μL. Subsequent dilutions and mixtures were based on the abundance intensity of the peptide standards compared to the corresponding peptides in the *S. aureus* cell sample. The dilutions and peptide standard concentrations are specified in the molecular ratio (Supplementary Table 2).

**LC-MS3 analysis quantitative mass spectrometry.** The sample mixtures were analyzed on an Orbitrap Lumos or Eclipse Tribrid™ mass spectrometer interfaced with an Easy-nLC1200 liquid chromatography system (Thermo Fisher Scientific). Peptides were trapped on an Acclaim Pepmap 100 C18 trap column (100 μm × 2 cm, particle size 5 μm, Thermo Fisher Scientific) and separated on an in-house packed analytical column (45 cm × 75 μm, particle size 3 μm, Reprosil-Pur C18, Dr. Maisch) using a stepped gradient from 4 to 80% acetonitrile in 0.2% formic acid over 90 min at a flow of 300 nL/min. The precursor ion mass spectra were acquired at a resolution of 120,000 and an *m/z* range of 375–1400. Using a cycle time of 3 s, the most abundant precursors with charges 2–7 were isolated with an *m/z* window of 0.7 and fragmented by collision-induced dissociation (CID) at 30%. Fragment spectra were recorded in the ion trap at Rapid scan rate. Dynamic exclusion was set to 45 s. The ten most abundant MS2 fragment ions were isolated using multi-notch isolation for further MS3 fragmentation. MS3 fragmentation was performed using higher-energy collision dissociation (HCD) at 55% and the MS3 spectra were recorded in the Orbitrap at 50 000 resolution and an *m/z* range of 100–500.

**Proteomic data analysis.** Data analysis was performed using Proteome Discoverer (ver 3.0, Thermo Fisher Scientific) The data were matched against *Staphylococcus aureus*, tax id 1280, SwissProt database (10245 entries), with the addition of Uniprot acc no Q8GNYF, using Sequest as a search engine with a precursor tolerance of 5 ppm and a fragment ion tolerance of 0.6 Da. Tryptic peptides were accepted with 0 missed cleavages. Methionine oxidation was set as a variable modification and TMTpro on lysine and peptide N-termini were set as fixed modifications. Fixed value was used for PSM validation with a threshold of 5%. For quantification TMT reporter ions were identified in the MS3 HCD spectra with 3 mmu

mass tolerance and the TMT reporter intensity values for each sample were exported as abundances for all peptides detected per protein. The SPS threshold was set to 40%, a Sequest HT threshold score of 2 was chosen. Only unique peptides were used for relative quantification. Protein molecular ratio in the *S. aureus* lysate were calculated based on the spiked protein concentration and the detected peptide abundances in Excel (Supplementary Data 1 and Supplementary Table 2). We report the mean value and the standard deviation between five individual mass spectrometry experiments based on the same *S. aureus* pellet (*n* = 5 technical replicates from *n* = 1 biological replicate).

**Di− and tripeptide formation assay**
The translation components used in the biochemical assays are from *E. coli* unless mentioned otherwise. His-tagged initiation factors (IF1, IF2, and IF3), elongation factors (EF-Tu, EF-Ts, and EF-G), 70S ribosome, XR7 mRNA with the coding sequence Met-Phe-Phe-Stop (AUG-UUC-UUU-UAA) and f[³H]Met-tRNA^fMet were prepared according to previous protocols[43,44]. The 70S initiation complex (IC) and elongation mixture (EM) were prepared separately by incubating the respective components in HEPES polymix (pH 7.5) buffer[43] at 37 °C for 15 min. The IC was assembled by combining 1 μM 70S ribosomes, 10 μM XR7 MFF mRNA, 1 μM f[³H]Met-tRNA^fMet, and 2 μM each of the three initiation factors (IF1, IF2, and IF3). 10 μM *S. aureus* FusB was added to IM to test its effect on di and tripeptide formation. The EM was prepared by mixing 10 μM EF-Tu, 10 μM EF-Ts, 10 μM tRNA^Phe, 0.2 mM Phe amino acid, 1 unit of Phe tRNA synthetase, and 2.5 μM *S. aureus* or *E. coli* EF-G (as indicated). Both mixtures were supplemented with an energy regeneration system comprising 1 mM GTP, 1 mM ATP, 10 mM phosphoenolpyruvate, 0.05 mg/mL pyruvate kinase, and 0.002 mg/mL myokinase. To capture the kinetics of peptide formation, equal volumes of the IM and EM were rapidly mixed in a Quench-Flow instrument (RQF-3, KinTek Corp.). The reactions were quenched by adding 17% formic acid at specific time points. Following quenching, peptides were released by KOH treatment. Mono-, di-, and tri-peptides were separated using a C18 reverse-phase column connected to a high-performance liquid chromatography system and detected with a β-RAM radioactive detector[43]. The di- and tripeptide formed were plotted against time and the rate of di/tripeptide formation was determined by fitting the data to a single exponential curve using Origin Pro 2016 software. This was done with two distinct experimental replicates (Supplementary Fig. 17) from which the means and standard deviations were calculated.

**Multiple sequence alignment**
The alignment was performed using Clustal Omega[45,46] at the EMBL-EBI web server[47] and displayed using Jalview[48].

**Analysis of interface areas**
Interface areas between EF-G and other components of the POST and FusB•EF-G•70S complexes were quantified using PDBePISA[49] based on structure coordinates truncated beyond 8 Å of EF-G.

**Figures**
Figures were rendered using UCSF ChimeraX[50], and assembled in Affinity Designer (Serif Europe Ltd, Nottinghamshire, UK).

**Reporting summary**
Further information on research design is available in the Nature Portfolio Reporting Summary linked to this article.

## Data availability
The cryo-EM maps and models in this study have been deposited in the Electron Microscopy Data Bank under the accession codes EMDB-51350 (CHI), EMDB-51351 (POST), EMDB-51352 (FusB•EF−G•70S), EMDB-51353 (FusB•EF−G•70S*), EMDB-51354 (FusB•70S:SSU), EMDB-

51355 (FusB•70S:LSU), EMDB-51356 (FusB•Sa70S:SSU), EMDB-51357 (FusB•Sa70S:LSU), and Protein Data Bank under the accession codes 9GHA (CHI), 9GHB (POST), 9GHC (FusB•EF-G•70S), 9GHD (FusB•EF-G•70S*), 9GHE (FusB•70S:SSU), 9GHF (FusB•70S:LSU), 9GHG (FusB•Sa70S:SSU), 9GHH (FusB•Sa70S:LSU). The mass spectrometry proteomics data have been deposited to the ProteomeXchange Consortium via the PRIDE[51] partner repository with the dataset identifier PXD062129. Source data are provided with this paper.

## Code availability

Custom Python scripts used during the analysis of the structures are available on Git Hub (https://github.com/adriangl97/pdb_python_tools).

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

## Acknowledgements

The authors thank Magnus Johansson for comments on the manuscript and on experimental design and Shirin Akbar for comments on the manuscript. The authors thank Anton Sabantcev for guidance regarding fluorescence polarization. The authors acknowledge the use of the Cryo-EM Uppsala facility for grid preparation and screening, funded by the Department of Cell and Molecular Biology, the Disciplinary Domains of Science and Technology, and of Medicine and Pharmacy at Uppsala University. Cryo-EM data were collected at the Cryo-EM Swedish National Facility funded by the Knut and Alice Wallenberg Foundation, the Erling-Persson Family Foundation, and the Kempe Foundation; SciLifeLab; Stockholm University; and Umeå University. This work benefited from access to Diamond Light Source and has been supported by iNEXT-Discovery project number 26535, funded by the Horizon 2020 program of the European Commission. We acknowledge access and support of the cryo-EM facilities at the UK national electron Bio-Imaging Centre (eBIC), proposal BI35989. The data processing was enabled by the Berzelius resource provided by the Knut and Alice Wallenberg Foundation at the National Supercomputer Centre. Proteomic analysis was performed at the Proteomics Core Facility, Sahlgrenska Academy, Gothenburg University, with financial support from SciLifeLab and BioMS (Vetenskapsrådet). This research was funded by grants from Uppsala Antibiotic Center to M.S., from the Swedish Research Council (2017-03827 and 2022-04511 to M.S.; 2023-05237, 2018-05946 and 2018-05498 to S.S.; 2016-06264 to M.S. and S.S.) and from the Knut and Alice Wallenberg Foundation (KAW 2017.0055) to S.S. A.G.L. has received fellowships from the Sven and Lilly Lawski Foundation and Kungl. Vetenskapsakademien.

## Author contributions

A.G.L. purified FusB and EF-G, prepared cryo-EM samples, processed cryo-EM data, modeled structures, performed and analyzed fluorescence polarization experiments, and grew *S. aureus* for mass spectrometry and ribosome purification. C.S.W. performed mass spectrometry experiments and analysis. A.G.L. and X.G. purified *S. aureus* ribosomes. X.G. and S.S. performed the di and tripeptide assays and purified the components. D.S.D.L. and M.S. supervised the work. A.G.L. and M.S. wrote the manuscript with help from all the authors. M.S. and S.S. secured funding.

## Funding

## Competing interests

The authors declare no competing interests.
