## [Transparent Peer Review file · Nature Communications]

Structural mechanism of FusB-mediated rescue from fusidic acid inhibition of protein synthesis

Corresponding Author: Professor Maria Selmer

Version 0:

Reviewer comments:

Reviewer #1

(Remarks to the Author)

General comments:

In this paper, Selmer and colleagues conducted a "time-resolved" Cryo-EM analysis using a clever method to elucidate how the drug resistance factor FusB resolves the EF-G - 70S ribosome complex trapped by fusidic acid (FA). Their efforts revealed that FusB binding induces structural rearrangement of EF-G, disrupting the interactions between the 70S ribosome and EF-G. This leads to dissociation of EF-G from the ribosome, rather than fusidic acid, resulting in the rescue of the FA-trapped ribosome. The molecular details they uncovered represent a unique mode of bacterial drug resistance acquisition and is also intriguing from the perspective of translation factor regulation. This information could be valuable to researchers across various fields.

However, regarding the structural interpretations mentioned in the latter part of the paper (FusB-LSU, FusB-SSU in Fig. 3), I find it difficult to accept the authors' hypothesis due to a lack of sufficient experimental evidence. If the authors are proposing the hypothesis that FusB inhibits the re-association of EF-G, further experimental analysis is required, such as mutational analysis of FusB or an evaluation of its binding affinities to the relevant factors. Detailed comments are provided below.

Major Critiques:

The authors report two structures of FusB bound to the ribosome independently of EF-G in Fig. 3. However, previous studies (reference-3 in the manuscript, PMID: 16390458) have reported that FusB binds to EF-G, but not to the ribosome. Given this discrepancy, I am concerned that the structures shown in Fig. 3 may be artifacts specific to the experimental conditions used in this study. Based on these structures, the authors argue that EF-G-independent FusB binding to the ribosome inhibits the re-association of EF-G after its dissociation. However, since EF-G is present in roughly equal quantity to ribosomes in the cell, FusB would preferentially bind to EF-G. Therefore, the FusB-70S complex the authors identified may exist only in very limited forms. Indeed, the results in Fig. 4 suggest that FusB does not inhibit the accommodation of aminoacyl-tRNA, implying that FusB interacts weakly or poorly with ribosome. Of course, the possibility that the structures in Fig. 3 indicate a previously unidentified function of FusB cannot be entirely ruled out. However, if the authors wish to discuss this hypothesis further in the paper, additional experiments, such as the following, would be necessary to substantiate it:

1. Mutational analysis of amino acid residues that contribute specifically to ribosomal binding (LSU, SSU) but not to EF-G interaction, to elucidate their role in fusidic acid resistance.
2. Evaluation of the affinity of FusB for the 70S ribosome or EF-G to determine whether the interaction between FusB and the 70S ribosome could compete with EF-G binding.

Other Critiques

Critiques-1: related to Fig. 1

1. The "time-resolved" Cryo-EM method used by the authors is intriguing and valuable. However, given that researchers may typically associate time-resolved Cryo-EM with "advanced instrumentation", it would be important to distinguish between these approaches to avoid potential confusion.
2. Has the reproducibility of this method been confirmed? While I understand that manual handling and the increased complexity of structural data make this challenging, it should be verified whether the disappearance of the FusB-EF-G-70S complex observed at 25 seconds (shown in Fig. 1) can be reproduced in another replicate or through biochemical assays under similar conditions.
3. In Fig. 1 or an accompanying figure, a schematic showing how time-resolved analysis was performed would help better convey the value and significance of this method.

4. The percentage of FusB-EF-G-70S particles over time, as depicted in Fig. 1, is crucial information. This data could be highlighted more prominently in the figure to emphasize its importance.

Critiques-2: related to Fig. 2

1. In the description related to Fig. 2 (lines 128-130), the authors mention that the contact sites between EF-G and the ribosome decrease from 41 to 23, but there is no figure illustrating this important finding. Supplementary Fig. 2 lacks specific explanations and is difficult to evaluate the individual interactions. A figure similar to Fig. 3, highlighting the major interactions, should be provided.
2. The presentation and use of color in Fig. 2 should be more organized. Particularly in 2c, the presence of FusB makes it difficult to observe domain III of EF-G.
3. Since the interactions are not completely lost, the description of "overall loss" (line-140 and elsewhere) is inappropriate and should be toned down.
4. Interactions mentioned elsewhere in the manuscript should also be visualized, at least as supplementary figures (e.g., line-123: K99-105, K174-K177).

Critiques-3: related to Fig. 3

1. In Fig. 3g, the symbols are too small and difficult to distinguish, so improvements are needed. Additionally, the regions mentioned in Fig. 3b-e should be indicated in Fig. 3g.
2. Although the manuscript discusses two structures, FusB-LSU and FusB-SSU, they are not clearly separated in the figure, which may confuse readers. It would be helpful to implement some visual distinctions to make the division between the two structures easier to understand.

Minor Points:

1. The residue numbers for the "lysine-rich loop" mentioned in Fig. 3 should be included in the text.
2. Regarding line 244, the text should mention the detailed concentration of FusB used in the study (10 μ M) and clarify the ratio of EF-G to FusB.
3. Lines 169-177: The type of experiment conducted in this section is not clearly described. Furthermore, while the interactions between FusB and the conserved residues of *E. coli* and *S. aureus* ribosomes are mentioned, no illustrations are provided. Some of representatives should be illustrated in the associated figure.
4. Lines 183 and 187: The term "early complex" should be clearly defined.
5. The notation "FusB•EF-G" and "FusB-EF-G" coexist. Please standardize these notations.
6. Line 221: Reference(s) should be provided.
7. The color scheme in Supplementary Fig. 6 is confusing and should be revised.
8. Lines 261-263: While this interpretation is accurate, adding some prospects for countermeasures could enhance the value of this paper.
9. Line 313: Revision is needed.
10. Line 635: Although the authors may have intentionally avoided specifying time points, approximate timing information, such as "early time point (6 s)," should be provided.

Reviewer #2

(Remarks to the Author)

Gonzalez-Lopez et al. provide a clear and compelling structural study revealing the mechanism for how FusB rescues bacterial ribosomes bound with elongation factor G arrested by the antibiotic fusidic acid. The work is technically sound, and the data support the conclusions drawn. The structural rearrangements seen in EF-G after FusB binding are quite interesting and provide yet another example for the power of extensive particle sorting in cryo-EM data sets. The FusB-70S complexes are unexpected although maybe indeed more of an experimental artifact as the authors suggest.

I have only minor comments to improve this manuscript:

Relevant to the discussion of FusB potentially competing with rebinding of EF-G, maybe the authors could comment in the discussion about the different interaction of FusB with EF-G in solution. For the tripeptide assay, the authors state that FusB has high affinity for *S. aureus* EF-G but does not bind to *E. coli* EF-G in solution (L168). Would that not affect any competition with rebinding on the ribosome?

L174: Supplementary Figure 4: it is impossible to see structural differences in an overall structure of the ribosome. Can the authors also provide a close-up view of FusB and ribosomal elements it interacts with?

L237: Provide the reference for the NMR-based model.

Methods: It would be useful to also provide the grid material for the cryo-EM grids used. Quantifoil grids could have been Cu, Ni or Au.

Typos:

Is there a reason why the author name of reference 11 is given all capitalized?

(Remarks to the Author)

The manuscript "Structural mechanism of FusB-mediated rescue from fusidic acid inhibition of protein synthesis" reports structure of the FusB with the FA-trapped EF-G on the ribosome, causing large-scale conformational changes of EF-G that break ribosome interactions. During protein synthesis, coupled movement of tRNAs and mRNA on the bacterial ribosome is catalyzed by the GTPase EF-G. To facilitate the successive addition of amino-acylated tRNA substrates, the resulting EF-G/GDP complex must dissociate from the ribosome, an event that is blocked by FA, leading to translational arrest and inhibition of bacterial growth. In vitro experiments revealed that FusB increased the rate of EF-G dissociation, indicating that it facilitates the conformational change needed for EF-G release and consequently accelerates the remobilization of FA-stalled ribosomes. However, the molecular mechanism of FusB-type resistance to fusidic acid has been a mystery since its discovery decades ago.

Here the authors report the structures of *S. aureus* FusB and EF-G bound to the *E. coli* 70S ribosome as determined by cryo-electron microscopy. Two structures displayed the identical conformation of the FA-trapped EF-G on the ribosome as reported by previous studies. Given the rapid rate at which FusB facilitates the dissociation of EF-G from the ribosome, the authors employed a method of mixing samples on grids. Within 6 seconds, the complex was frozen to capture the structure of FusB bound to the 70S ribosome. Additionally, the interaction between FusB and EF-G resulted in the displacement of EF-G from its original position. The authors also reported several seemingly non-specific structures of FusB or FusB/EF-G bound to the 70S ribosome.

This work is crucial for understanding the mechanism by which FusB promotes the dissociation of EF-G. However, I have several major concerns.

Major concerns.

1. Dataset 1 and 2 from 10 s were collected from different position on a single grid and showed strikingly different populations of complexes, which indicated a diffusion-dependent gradient of ribosomal states (line 81). However, in Figure 1, the authors used changes in the proportions of different states to illustrate the reaction process of FusB at 6 and 25 s time points (lines 95-104). Can the authors ensure that the differences between the 6s and 25s data are not influenced by the positions on the grids? Given the strong emphasis the authors placed on these numbers, it is crucial to demonstrate that the numbers indeed reflect the action process of FusB accurately. Several specific questions about these numbers are listed below:
 - a. The numbers of "CHI" plus "POST" are 22.5% in dataset 2 (10 s), 5.0% in dataset 1 (10 s), 12.7% in early (6 s) and 2.7% in 25 s sample, which implies that the sequence of reaction is dataset 2, 6 s, dataset 1, and 25s. Fitting to this sequence, the proportions of FusB•EF-G•70S are 1.6%, 1.6%, 4.4%, and 0%. Please explain the relatively high proportion of FusB•EF-G•70S (4.4%) observed in dataset 1. According to the authors' interpretation, this percentage should progressively decrease.
 - b. Why do the proportions of the other three complexes show no discernible pattern across different datasets? Especially the FusB•70S:SSU complex in dataset 2, the number is only 1.1%. If these numbers are not indicative of anything significant, then the ones for FusB•EF-G•70S also lack persuasion.
 - c. Line 189-191: "Since the proportion of FusB•EF-G•70S* is the same at both timepoints, this more likely represents a rebinding event of FusB•EF-G." Don't forget the numbers are 2.5% and 1.8% in dataset 1 and 2, respectively.
2. I still have not grasped the function of FusB•70S:SSU and FusB•70S:LSU. Are they just unspecific binding to the empty A-site? Why they are important?
3. Here is the critical thing: the conformations of FusB•EF-G in FusB•EF-G•70S and FusB•EF-G•70S* are the same, just have different orientation. Why FusB•EF-G•70S* is rebinding, while FusB•EF-G•70S is the real "dissociating" intermediate. Is it possible FusB•EF-G•70S is also a "rebinding" event? The number 0% in 25 s is just due to the fact that this small proportion does not stably exist.
4. Is there any P-site tRNA in "A-site tRNA" and "empty A-site" conformation (Fig.S9)? How are these two states generated? Why is the proportion of the empty A-site particularly high, especially given that the authors reported FusB has a strong propensity for ribosome binding, independently of EF-G?
5. Why the authors used lowpass filtered density to represent the map quality with coordinates? Indicated the exact resolution used for filtering.
6. Line 241, why "the FusB-EF-G interface observed in our structures will be retained in solution"?

Minor concerns.

1. Keep all as "FusB•EF-G".
2. Please show the density for Fig. S1C.
3. In Fig. S2C, domain II does not seem to fit properly. Same for domain IV in Fig. S2B.
4. Are the alignments in lines 117 and 126 the same? The sentences are repeated.
5. Remove one of the EF-G in keywords.
6. Remove "nm" after OD600 in line 281.
7. Line 53: What is the "wild-type" refer to?
8. Line 62: Why "FusB also increases the turnover of *S. aureus* EF-G on the ribosome in absence of FA22"? Explanations could be added in the Discussion.
9. Can the complexes on *E. coli* 70S explain the mechanism on *S. aureus* 70S? Add some discussion about this.
10. Line 181: Should be "Domain V of EF-G". Label uL6 in the figure.

Reviewer comments:

Reviewer #1

(Remarks to the Author)

General comments:

I appreciate the authors' sincere efforts in addressing the comments through additional experiments and revisions to the manuscript. The newly provided experimental data regarding FusB's binding to the 70S ribosome have alleviated most of my concerns, allowing me to accept the authors' claims.

However, regarding the newly added data, I believe the Discussion section should be further expanded to include a more detailed explanation of the dissociation constants (K_D) and molecular ratios among EF-G, ribosomes, and FusB.

1. The discussion should incorporate past studies (PBID: 22308410) along with the current experimental results related to new Fig. 5a-b and previously measured K_D values between FusB and EF-G, as well as between FusB and the 70S ribosome.

2. The authors should explicitly state that their analysis does not clarify whether FusB preferentially binds to EF-G or the 70S ribosome under conditions where both coexist.

3. Based on a previous report (PBID: 16390458), it is likely that FusB preferentially binds to EF-G rather than the ribosome, and this should be explicitly mentioned in the text.

4. Since excessive interactions between FusB and EF-G could negatively impact protein synthesis, the manuscript should better clarify how FusB is regulated by leader peptide-mediated translation control.

Additionally, a more detailed discussion (e.g., on FusB homologs unrelated to antibiotic resistance) would make it easier to understand the potential function of FusB's independent binding to the 70S ribosome beyond fusidic acid resistance.

Without such discussion, while the formation of the FusB-70S complex itself may be convincing, it remains difficult to accept that this phenomenon holds physiological significance and occurs in actual cellular conditions.

Furthermore, Supplementary Table 2 presents critical data on the ratio of FusB to EF-G, but the legend and explanations are insufficient, making it difficult to interpret the validity of the results. Additionally, MS data deposition is not mentioned in the Data Availability section. The details of the MS analysis, including which peptide fragments were used for comparative quantification of FusB and EF-G, should be explicitly provided.

Minor Comments:

Line 154-155:

Since there is no experimental data other than FusB, the expression should be toned down slightly.

Line 165-166:

A past reference should be cited here, and the K_D value of FusB-EF-G should be briefly mentioned.

Line 186-188:

It should be explicitly stated that the experiment was conducted in the absence of fusidic acid and EF-G.

Line 262:

There is no explanation indicating that (~63 nM) represents a K_D value.

References 13, 28, 41, 42, 49:

The formatting appears to be incorrect; please check for any issues.

Supplementary Reference-3:

Ensure that the formatting of this reference is correct.

Reviewer #3

(Remarks to the Author)

In the revised manuscript, Gonzalez-Lopez et al. present a clearer mechanism detailing how FusB rescues bacterial ribosomes that have been stalled by the antibiotic fusidic acid in the presence of elongation factor G. The data now substantiate the conclusions drawn.

Answers to reviewers

For clarity, all answers are written in green.

Reviewer 1:

General comments:

In this paper, Selmer and colleagues conducted a "time-resolved" Cryo-EM analysis using a clever method to elucidate how the drug resistance factor FusB resolves the EF-G - 70S ribosome complex trapped by fusidic acid (FA). Their efforts revealed that FusB binding induces structural rearrangement of EF-G, disrupting the interactions between the 70S ribosome and EF-G. This leads to dissociation of EF-G from the ribosome, rather than fusidic acid, resulting in the rescue of the FA-trapped ribosome. The molecular details they uncovered represent a unique mode of bacterial drug resistance acquisition and is also intriguing from the perspective of translation factor regulation. This information could be valuable to researchers across various fields.

However, regarding the structural interpretations mentioned in the latter part of the paper (FusB-LSU, FusB-SSU in Fig. 3), I find it difficult to accept the authors' hypothesis due to a lack of sufficient experimental evidence. If the authors are proposing the hypothesis that FusB inhibits the re-association of EF-G, further experimental analysis is required, such as mutational analysis of FusB or an evaluation of its binding affinities to the relevant factors. Detailed comments are provided below.

Major Critiques:

The authors report two structures of FusB bound to the ribosome independently of EF-G in Fig. 3. However, previous studies (reference-3 in the manuscript, PMID: 16390458) have reported that FusB binds to EF-G, but not to the ribosome. Given this discrepancy, I am concerned that the structures shown in Fig. 3 may be artifacts specific to the experimental conditions used in this study. Based on these structures, the authors argue that EF-G-independent FusB binding to the ribosome inhibits the re-association of EF-G after its dissociation. However, since EF-G is present in roughly equal quantity to ribosomes in the cell, FusB would preferentially bind to EF-G. Therefore, the FusB-70S complex the authors identified may exist only in very limited forms. Indeed, the results in Fig. 4 suggest that FusB does not inhibit the accommodation of aminoacyl-tRNA, implying that FusB interacts weakly or poorly with ribosome. Of course, the possibility that the structures in Fig. 3 indicate a previously unidentified function of FusB cannot be entirely ruled out. However, if the authors wish to discuss this hypothesis further in the paper, additional experiments, such as the following, would be necessary to substantiate it:

1. Mutational analysis of amino acid residues that contribute specifically to ribosomal binding (LSU, SSU) but not to EF-G interaction, to elucidate their role in fusidic acid resistance.

2. Evaluation of the affinity of FusB for the 70S ribosome or EF-G to determine whether the interaction between FusB and the 70S ribosome could compete with EF-G binding.

Thanks for pointing this out. Unfortunately, mutational analysis is very challenging, as most residues involved in interactions with the ribosome (in FusB•SSU and FusB•LSU) are also involved in interactions with EF-G and the ribosome (in FusB•EF-G•70S), see Figure 4.

To address this question, we have measured the affinity of fluorescein-labeled FusB to the ribosome by fluorescence polarization, resulting in K_D of ~320 nM to the 70S ribosome (Figure 5a-b). This seems to agree with what we observe in the cryo-EM data.

In addition, we have determined the *in vivo* relative concentration of FusB to EF-G in the clinical *S. aureus* strain harboring the pUB101 plasmid encoding for FusB, using mass spectrometry. The EF-G:FusB molar ratio of 2.3 ± 0.3 shows that FusB is present at low micromolar concentration.

Together, our additional data supports the idea that FusB can bind to ribosomes *in vivo*, but the function of this interaction remains unknown, and we removed the speculation that this has a putative role in FA resistance.

Other Critiques

Critiques-1: related to Fig. 1

1. The "time-resolved" Cryo-EM method used by the authors is intriguing and valuable. However, given that researchers may typically associate time-resolved Cryo-EM with "advanced instrumentation", it would be important to distinguish between these approaches to avoid potential confusion.

Thank you for your comment. We hope that the new figure, Supplementary Fig. 1, makes the distinction clearer. For clarity, we also gave this method a name, the "mix-on-grid method".

2. Has the reproducibility of this method been confirmed? While I understand that manual handling and the increased complexity of structural data make this challenging, it should be verified whether the disappearance of the FusB-EF-G-70S complex observed at 25 seconds (shown in Fig. 1) can be reproduced in another replicate or through biochemical assays under similar conditions.

Thank you for your comment. We are not sure of the exact detection limit of our cryo-EM processing approach, but have changed the previous "0 %" to "ND (not detected)" in Figure 1. However, we think that the 25 s timepoint should be reproducible, as the samples were prepared by conventional mixing in a tube with slightly larger volumes. The main challenge of this work has rather been to be able to visualize this short-lived state, which required keeping all components on ice and mixing on the grid. At 37 °C, we expect the process to be 8 to 10-fold faster, and we consider the full biochemical characterization of this process or repetition of the cryo-EM experiment to be beyond the scope of this manuscript.

3. In Fig. 1 or an accompanying figure, a schematic showing how time-resolved analysis was performed would help better convey the value and significance of this method.

Thank you for the suggestion. We have added a schematic of the mix-on-grid method as Supplementary Figure 1.

4. The percentage of FusB-EF-G-70S particles over time, as depicted in Fig. 1, is crucial information. This data could be highlighted more prominently in the figure to emphasize its importance.

Thanks for the suggestion. We have increased the font size of the numbers in Figure 1 to improve its visibility.

Critiques-2: related to Fig. 2

1. In the description related to Fig. 2 (lines 128-130), the authors mention that the contact sites between EF-G and the ribosome decrease from 41 to 23, but there is no figure illustrating this important finding. Supplementary Fig. 2 lacks specific explanations and is difficult to evaluate the individual interactions. A figure similar to Fig. 3, highlighting the major interactions, should be provided.

Thanks for the suggestion. We have removed this metric and instead, to provide more quantitative data, calculated the interface area between EF-G and the ribosome using PDBePISA (Supplementary Table 1). We have also added Supplementary Figure 9 that highlights some of the most prominent differences.

2. The presentation and use of color in Fig. 2 should be more organized. Particularly in 2c, the presence of FusB makes it difficult to observe domain III of EF-G.

Thank you for noticing this. We have changed the color of domain III to make it more visible and we have modified the labels to improve the clarity of presentation.

3. Since the interactions are not completely lost, the description of "overall loss" (line-140 and elsewhere) is inappropriate and should be toned down.

Thanks for pointing this out, we have modified the text accordingly.

4. Interactions mentioned elsewhere in the manuscript should also be visualized, at least as supplementary figures (e.g., line-123: K99-105, K174-K177).

Thank you for the suggestion. We have added this to Supplementary Figure 8e-f.

Critiques-3: related to Fig. 3

1. In Fig. 3g, the symbols are too small and difficult to distinguish, so improvements are needed. Additionally, the regions mentioned in Fig. 3b-e should be indicated in Fig. 3g.

Thanks for pointing this out. We have made our best to improve the symbols and clarity in the MSA figure (now Figure 4). For more clarity we have also separated FusB•SSU and FusB•LSU interactions and moved the MSA into a separate figure. We consider that additional labels would clutter Fig. 3g (now Figure 4), but we hope that the changes allow the reader to easily locate these regions.

2. Although the manuscript discusses two structures, FusB-LSU and FusB-SSU, they are not clearly separated in the figure, which may confuse readers. It would be helpful to implement some visual distinctions to make the division between the two structures easier to understand.

Thank you for your comment. We now combine an overview figure of each structure (Figure 3a,d) with zoomed in figures for specific regions and for clarity have added additional labels.

Minor Points:

1. The residue numbers for the "lysine-rich loop" mentioned in Fig. 3 should be included in the text.

Thanks for noticing, we have done as suggested.

2. Regarding line 244, the text should mention the detailed concentration of FusB used in the study (10 μ M) and clarify the ratio of EF-G to FusB.

Thank you for your suggestion, we have added the requested information.

3. Lines 169-177: The type of experiment conducted in this section is not clearly described. Furthermore, while the interactions between FusB and the conserved residues of E. coli and S. aureus ribosomes are mentioned, no illustrations are provided. Some of representatives should be illustrated in the associated figure.

Thanks for pointing this out. We have now clarified that this is done by cryo-EM and added some representative illustrations (Supplementary Figure 12b-c, e-f).

4. Lines 183 and 187: The term "early complex" should be clearly defined.

Thank you for pointing this out. We have replaced "early complex" with FusB•EF-G•70S throughout the manuscript.

5. The notation "FusB•EF-G" and "FusB-EF-G" coexist. Please standardize these notations.

Thank you for noticing. It should now be consistent, we only use "FusB-EF-G" when grammatically required.

6. Line 221: Reference(s) should be provided.

We have added the reference.

7. The color scheme in Supplementary Fig. 6 is confusing and should be revised.

Thanks for pointing this out. We have changed the color of the NMR structure to make it more consistent with the rest of the figures.

8. Lines 261-263: While this interpretation is accurate, adding some prospects for countermeasures could enhance the value of this paper.

Thank you for your suggestion. We have added some possibilities for countermeasures.

9. Line 313: Revision is needed.

Thank you, it is now corrected.

10. Line 635: Although the authors may have intentionally avoided specifying time points, approximate timing information, such as "early time point (6 s)," should be provided.

We agree with your comment and have added this to the figures.

Reviewer 2:

Gonzalez-Lopez et al. provide a clear and compelling structural study revealing the mechanism for how FusB rescues bacterial ribosomes bound with elongation factor G arrested by the antibiotic fusidic acid. The work is technically sound, and the data support the conclusions drawn. The structural rearrangements seen in EF-G after FusB binding are quite interesting and provide yet another example for the power of extensive particle sorting in cryo-EM data sets. The FusB-70S complexes are unexpected although maybe indeed more of an experimental artifact as the authors suggest.

I have only minor comments to improve this manuscript:

Relevant to the discussion of FusB potentially competing with rebinding of EF-G, maybe the authors could comment in the discussion about the different interaction of FusB with EF-G in solution. For the tripeptide assay, the authors state that FusB has high affinity for *S. aureus* EF-G but does not bind to *E. coli* EF-G in solution (L168). Would that not affect any competition with rebinding on the ribosome?

Thank you for your comment. We have expanded this part of the discussion, also incorporating additional data on FusB-70S affinity and *in vivo* concentration of FusB.

For the tripeptide assay, we used *E. coli* EF-G as a control to remove the effect of FusB binding to EF-G and not the ribosome. We agree that the concentration of free EF-G and free FusB will be different in those two experiments, depending on whether the two proteins bind to each other or not, but there is no sign of FusB-mediated inhibition of ternary complex binding (in dipeptide formation) in any of the two experiments. In line with suggestions from other referees we have removed the suggestion that FusB binding to 70S may prevent re-binding of EF-G.

L174: Supplementary Figure 4: it is impossible to see structural differences in an overall structure of the ribosome. Can the authors also provide a close-up view of FusB and ribosomal elements it interacts with?

Thank you for your suggestion. We have added additional panels to the figure (now Supplementary Figure 12) that compare the specific interactions shown in Figure 3.

L237: Provide the reference for the NMR-based model.

Thanks for noticing this mistake, it has been added.

Methods: It would be useful to also provide the grid material for the cryo-EM grids used. Quantifoil grids could have been Cu, Ni or Au.

Thanks for the suggestion, we have added this information.

Typos:

Is there a reason why the author name of reference 11 is given all capitalized?

Thanks for noticing, this was a typo and has been corrected.

Reviewer 3:

The manuscript “Structural mechanism of FusB-mediated rescue from fusidic acid inhibition of protein synthesis” reports structure of the FusB with the FA-trapped EF-G on the ribosome, causing large-scale conformational changes of EF-G that break ribosome interactions. During protein synthesis, coupled movement of tRNAs and mRNA on the bacterial ribosome is catalyzed by the GTPase EF-G. To facilitate the successive addition of amino-acylated tRNA substrates, the resulting EF-G/GDP complex must dissociate from the ribosome, an event that is blocked by FA, leading to translational arrest and inhibition of bacterial growth. In vitro experiments revealed that FusB increased the rate of EF-G dissociation, indicating that it facilitates the conformational change needed for EF-G release and consequently accelerates the remobilization of FA-stalled ribosomes. However, the molecular mechanism of FusB-type resistance to fusidic acid has been a mystery since its discovery decades ago.

Here the authors report the structures of *S. aureus* FusB and EF-G bound to the *E. coli* 70S ribosome as determined by cryo-electron microscopy. Two structures displayed the identical conformation of the FA-trapped EF-G on the ribosome as reported by previous studies. Given the rapid rate at which FusB facilitates the dissociation of EF-G from the ribosome, the authors employed a method of mixing samples on grids. Within 6 seconds, the complex was frozen to capture the structure of FusB bound to the 70S ribosome. Additionally, the interaction between FusB and EF-G resulted in the displacement of EF-G from its original position. The authors also

reported several seemingly non-specific structures of FusB or FusB/EF-G bound to the 70S ribosome.

This work is crucial for understanding the mechanism by which FusB promotes the dissociation of EF-G. However, I have several major concerns.

Major concerns.

1. Dataset 1 and 2 from 10 s were collected from different position on a single grid and showed strikingly different populations of complexes, which indicated a diffusion-dependent gradient of ribosomal states (line 81). However, in Figure 1, the authors used changes in the proportions of different states to illustrate the reaction process of FusB at 6 and 25 s time points (lines 95-104). Can the authors ensure that the differences between the 6s and 25s data are not influenced by the positions on the grids? Given the strong emphasis the authors placed on these numbers, it is crucial to demonstrate that the numbers indeed reflect the action process of FusB accurately. Several specific questions about these numbers are listed below:

a. The numbers of “CHI” plus “POST” are 22.5% in dataset 2 (10 s), 5.0% in dataset 1 (10 s), 12.7% in early (6 s) and 2.7% in 25 s sample, which implies that the sequence of reaction is dataset 2, 6 s, dataset 1, and 25s. Fitting to this sequence, the proportions of FusB•EF-G•70S are 1.6%, 1.6%, 4.4%, and 0%. Please explain the relatively high proportion of FusB•EF-G•70S (4.4%) observed in dataset 1. According to the authors’ interpretation, this percentage should progressively decrease.

Thanks for pointing this out. The comment made us realize that we were not sufficiently clear in explaining that datasets 1 and 2 are smaller preliminary datasets and for this reason have limitations as input to advanced particle classification, compared to the high-resolution datasets used to produce the presented structures. The grid for datasets 1 and 2 unfortunately seemed to have no continuous carbon support, as we observed a higher degree of preferred orientation in both datasets. In addition, one of those datasets is from a Glacios microscope with Falcon III camera, which limits the resolving power of the performed 3D classifications. Thanks to your comments we now realize that we should not put weight on the specific percentages for those two datasets. We included those to support our initial observation of a striking difference in the population of FusB•70S and EF-G•70S on different parts of the same grid, indicating incomplete mixing/diffusion of FusB. These two states show double digit differences in population, despite the limitations of those datasets, prompting us to make use of the mix-on-grid method. However, we do not judge small differences in particle populations between these preliminary datasets as trustworthy enough to draw mechanistic conclusions.

Furthermore, the concentrations of components used to prepare grids for the preliminary and high-resolution datasets were different, as discussed in the manuscript (details in the methods section), making the datasets not directly comparable. To not mislead the reader to overinterpret

the comparison between the preliminary datasets, we have removed the previous Extended Data Figure 1 and now only provide the processing workflows in Supplementary Figures 2 and 3 to support our preliminary analysis.

b. Why do the proportions of the other three complexes show no discernible pattern across different datasets? Especially the FusB•70S:SSU complex in dataset 2, the number is only 1.1%. If these numbers are not indicative of anything significant, then the ones for FusB•EF-G•70S also lack persuasion.

Thank you for your comment. As described above, the preliminary datasets, datasets 1 and 2, cannot be directly compared to the high-resolution datasets, early (6s) and 25 s, due to preferred particle orientation and different concentration of the components.

c. Line 189-191: “Since the proportion of FusB•EF-G•70S* is the same at both timepoints, this more likely represents a rebinding event of FusB•EF-G.” Don’t forget the numbers are 2.5% and 1.8% in dataset 1 and 2, respectively.

Thank you for your comment. As described above, we do not judge the small differences in particle populations between these preliminary datasets as trustworthy enough to draw mechanistic conclusions.

2. I still have not grasped the function of FusB•70S:SSU and FusB•70S:LSU. Are they just unspecific binding to the empty A-site? Why they are important?

Thanks for pointing this out. Based on the presented results, we cannot tell why FusB binds directly to the ribosome. However, in response to comments from reviewer 1, we now show that binding is likely to occur at physiological concentrations of FusB and the translation machinery. The new affinity measurements with fluorescence polarization and relative quantitative mass spectrometry data have been added to the revised manuscript (Figure 5, Supplementary Table 2). We removed the speculation that this binding can prevent re-binding of EF-G with fusidic acid. Future studies will elucidate the function of the direct interaction between FusB and the ribosomal A site.

3. Here is the critical thing: the conformations of FusB•EF-G in FusB•EF-G•70S and FusB•EF-G•70S* are the same, just have different orientation. Why FusB•EF-G•70S* is rebinding, while FusB•EF-G•70S is the real “dissociating” intermediate. Is it possible FusB•EF-G•70S is also a “rebinding” event? The number 0% in 25 s is just due to the fact that this small proportion does not stably exist.

Thanks for pointing out this unclarity. We base this reasoning on knowledge of the structure of the FA-inhibited state of EF-G on the ribosome (this work and several previous papers). Figure 2 and Supplementary Video 1 shows that the binding-site of EF-G to the ribosome in FusB•EF-G•70S is closely related to the binding site in the POST state, but that binding of FusB induces conformational changes in EF-G that reduce its contacts with the ribosome. The presence of clear

density for FA in the FusB•EF-G•70S complex proves that this is a true rescue complex, since FA only binds to EF-G on the ribosome after GTP hydrolysis.

In the FusB•EF-G•70S* complex, EF-G makes non-canonical interactions with the ribosome, never observed before, and there is no clear link between this binding site and the functional binding of EF-G to the ribosome. Further, there is no density for FA, and our conclusion is that this state can only result from re-binding of the FusB-EF-G complex to the ribosome.

4. Is there any P-site tRNA in “A-site tRNA” and “empty A-site” conformation (Fig.S9)? How are these two states generated? Why is the proportion of the empty A-site particularly high, especially given that the authors reported FusB has a strong propensity for ribosome binding, independently of EF-G?

Thanks for asking this. Our complex consists of *E. coli* 70S ribosomes with a short mRNA, tRNA^{fMet} bound to a P-site AUG codon, plus EF-G, GTP and FA, to which FusB is added. During the experiment, most ribosomes on the grid are rescued from FA inhibition, which explains the high proportion of empty A-site. The P-site is still occupied by tRNA^{fMet} in these reconstructions. There is no cognate A-site tRNA present in the mixture, but tRNA^{fMet} shows unspecific binding to the vacant A-site.

FusB can bind to both the free excess EF-G and to the classical-state 70S ribosome, with ~5-fold lower affinity to the 70S ribosome, which explains why we do not observe even higher occupancy of FusB. At the concentrations used in grid preparation (5 μM each of EF-G and FusB, 0.5 of 70S), most of the FusB molecules are likely bound to EF-G. For these reasons, it is not surprising that most ribosomes have an empty A site at conditions of rescue.

5. Why the authors used lowpass filtered density to represent the map quality with coordinates? Indicated the exact resolution used for filtering.

Thank you for your comment. We used the local filtering algorithm from cryoSPARC, which sharpens the map and then lowpass filters locally according to the estimated local resolution. We decided to use this representation to prevent over-interpretation of lower-resolution regions of the map, while showing the presence of higher-resolution features in the higher-resolution regions. For this reason, we cannot specify a specific resolution for lowpass filtering, which follows the local resolution estimation (Supplementary Figure 4).

6. Line 241, why “the FusB-EF-G interface observed in our structures will be retained in solution”?

Thanks for pointing out this unclarity. FusB and EF-G form a high-affinity complex in solution (Cox *et al.*, 2012). Based on our structure, which does not agree with a previous NMR-based model (Supplementary Figure 14), we expect that the main FusB-EF-G interface remains the same after release from the ribosome, although the presence of FusB in combination with the absent ribosome contacts makes EF-G more dynamic.

Minor concerns.

1. Keep all as “FusB•EF-G”.

Thank you for noticing this error, it is now changed unless grammatically required.

2. Please show the density for Fig. S1C.

Thank you for the suggestion, we have added this (now Supplementary Figure 7d).

3. In Fig. S2C, domain II does not seem to fit properly. Same for domain IV in Fig. S2B.

Thanks for pointing this out. Some regions of EF-G show poor density in this map, likely disordered due to the destabilization caused by the loss of contact with the ribosome. However, the map allowed confident fitting of the whole domains.

4. Are the alignments in lines 117 and 126 the same? The sentences are repeated.

Thank you for noticing this. We have rewritten this part to increase clarity.

5. Remove one of the EF-G in keywords.

Thank you, it has been removed.

6. Remove “nm” after OD600 in line 281.

Thank you, we removed it.

7. Line 53: What is the “wild-type” refer to?

Thank you for noticing, we changed this to “susceptible strains”, which is what we meant.

8. Line 62: Why “FusB also increases the turnover of *S. aureus* EF-G on the ribosome in absence of FA22”? Explanations could be added in the Discussion.

Thanks for the suggestion. Experiments in Cox *et al.* 2012 show that FusB increases the turnover of EF-G in a fluorescence-based assay monitoring EF-G dissociation. Based on our results, it is possible that FusB can bind to EF-G on the ribosome also in absence of FA, but because of the regulation of expression of FusB by translational attenuation, this is unlikely to happen *in vivo*. This has been added to the discussion.

9. Can the complexes on *E. coli* 70S explain the mechanism on *S. aureus* 70S? Add some discussion about this.

Thank you for your comment. It has previously been shown that the heterogeneous system with *E. coli* ribosomes and *S. aureus* EF-G works well to study FusB-mediated rescue (Guo *et al.*, 2012, Cox *et al.*, 2012). We have validated some of the results with *S. aureus* ribosomes, such as direct binding of FusB to the 70S ribosome (Figure 5, Supplementary Figure 12). We have also added this to the discussion.

10. Line 181: Should be “Domain V of EF-G”. Label uL6 in the figure.

Thank you for your comment. We have specified that it is domain V that contacts those regions. and labeled uL6 as suggested.

REVIEWERS' COMMENTS

Reviewer #1 (Remarks to the Author):

General comments:

I appreciate the authors' sincere efforts in addressing the comments through additional experiments and revisions to the manuscript. The newly provided experimental data regarding FusB's binding to the 70S ribosome have alleviated most of my concerns, allowing me to accept the authors' claims.

However, regarding the newly added data, I believe the Discussion section should be further expanded to include a more detailed explanation of the dissociation constants (K_D) and molecular ratios among EF-G, ribosomes, and FusB.

Thanks, the comment is important for clarity, and we have added a more extensive discussion.

1. The discussion should incorporate past studies (PBIID: 22308410) along with the current experimental results related to new Fig. 5a-b and previously measured K_D values between FusB and EF-G, as well as between FusB and the 70S ribosome.

Thanks, this has been added.

2. The authors should explicitly state that their analysis does not clarify whether FusB preferentially binds to EF-G or the 70S ribosome under conditions where both coexist.

Yes, this has been added.

3. Based on a previous report (PBIID: 16390458), it is likely that FusB preferentially binds to EF-G rather than the ribosome, and this should be explicitly mentioned in the text.

Thanks. The measured dissociation constants in combination with the concentrations of EF-G and classical-state ribosomes with empty A site does suggest preferential binding to EF-G. We have done our best to clarify this in the manuscript.

4. Since excessive interactions between FusB and EF-G could negatively impact protein synthesis, the manuscript should better clarify how FusB is regulated by leader peptide-mediated translation control.

Thanks, this has been added.

Additionally, a more detailed discussion (e.g., on FusB homologs unrelated to antibiotic resistance) would make it easier to understand the potential function of FusB's independent binding to the 70S ribosome beyond fusidic acid resistance. Without such discussion, while the formation of the FusB-70S complex itself may be convincing, it remains difficult to accept that this phenomenon holds physiological significance and occurs in actual cellular conditions.

Thanks, we have added a few sentences, also highlighting that there are several prior examples of antibiotic resistance proteins that have evolved from housekeeping proteins.

Furthermore, Supplementary Table 2 presents critical data on the ratio of FusB to EF-G, but the legend and explanations are insufficient, making it difficult to interpret the validity of the results. Additionally, MS data deposition is not mentioned in the Data Availability section. The details of the MS analysis, including which peptide fragments were used for comparative quantification of FusB and EF-G, should be explicitly provided.

Thanks for pointing this out. An expanded and clarifying legend has been added to Supplementary Table 2 and Supplementary Data 1 provides mass spectrometry results per detected peptide for the relative quantification of FusB. Mass spectrometry data have been deposited in the ProteomeXchange Consortium via the PRIDE partner repository, and this is now specified under Data Availability.

Minor Comments:

Line 154-155:

Since there is no experimental data other than FusB, the expression should be toned down slightly.

We re-phrased the statement as "suggesting that these proteins would also be able to bind to the ribosome similarly"

Line 165-166:

A past reference should be cited here, and the K_D value of FusB-EF-G should be briefly mentioned.

Thanks. Since this is the results section, describing our experiments and results regarding direct binding of FusB to the 70S ribosome, we don't find it appropriate to add the K_D value and the reference to published affinity between FusB and EF-G. However, as described above, we have added a more extensive discussion of this in the discussion section.

Line 186-188:

It should be explicitly stated that the experiment was conducted in the absence of fusidic acid and EF-G.

Thanks for pointing this out, this has been clarified.

Line 262:

There is no explanation indicating that (~63 nM) represents a K_D value.
Thanks for pointing this out, we have clarified that this is the K_D value.

References 13, 28, 41, 42, 49:

The formatting appears to be incorrect; please check for any issues.
This has been corrected.

Supplementary Reference-3:

Ensure that the formatting of this reference is correct.
This has been corrected.

Reviewer #3 (Remarks to the Author):

In the revised manuscript, Gonzalez-Lopez et al. present a clearer mechanism detailing how FusB rescues bacterial ribosomes that have been stalled by the antibiotic fusidic acid in the presence of elongation factor G. The data now substantiate the conclusions drawn.

Thanks!